# Incorporating Importance Weighting in Optimal Transport Based Domain Alignment

Okan Koç [1]   Alexander Soen [2]   Shanglin Li [3]   Masashi Sugiyama [1 4]

## Abstract

Domain adaptation theory studies upper bounds on the target risk in order to mitigate performance loss of machine learning models due to distribution shift. In this paper, we take a closer look at the optimization of one such bound based on optimal transport (OT) and propose various strategies that improve the optimization in practice. We first introduce *gradual shift* and *probabilistic margin* assumptions to control the incomputable entanglement term that appears in the bounds. We prove that under these assumptions, better optimization of the computable part of the bound can translate to better target accuracies. Motivated by this fact, we tighten the bound, via importance weighting of the source (output) distribution, to obtain the *weighted* Wasserstein regularized risk ($W^2R^2$), that is often easier to minimize than the original bound. $W^2R^2$ is shown to be equivalent to an unbalanced OT problem, which in the limit converges to a nearest neighbor based alignment strategy. We highlight the tradeoffs faced with such an approach and show that a suitably regularized $W^2R^2$ improves over the state of the art and is robust to multiple distribution shifts under different models, confirming, moreover, the validity of our assumptions.

## 1. Introduction

Deep learning is a powerful and effective tool to use in the standard supervised learning setting (Bengio et al., 2017), where it is assumed that the labeled source (training) data

[1]Center For Advanced Intelligence Project, RIKEN, Tokyo, Japan [2]Department of Information Science and Engineering, KTH Royal Institute of Technology, Stockholm, Sweden [3]BIFOLD, Berlin Institute for the Foundations of Learning and Data, Berlin, Germany [4]Dept. of Complexity Science and Engineering, The University of Tokyo, Tokyo, Japan. Correspondence to: Okan Koç <okan.koc@riken.jp>.

*Proceedings of the 43rd International Conference on Machine Learning*, Seoul, South Korea. PMLR 306, 2026. Copyright 2026 by the author(s).

and the unlabeled target (test) data are drawn from the same underlying data distribution. In such a scenario, it is possible to make guarantees for models trained via empirical risk minimization (ERM) (Shalev-Shwartz & Ben-David, 2014). Unfortunately, deep learning has shown to be quite brittle under distribution shifts (Hendrycks & Dietterich, 2019), where the data distribution used to train a model is different from the distribution that the model is exposed to at inference time. Ultimately, this often prevents wider deployment of these trained models.

As a result of this lack of robustness, one of the main research directions in the field of domain adaptation (DA) has been finding *invariant representations*, often via domain alignment techniques (Ganin et al., 2016; Courty et al., 2017). Such methods only require unlabeled target data to define an additional regularization term (typically a divergence) added to source risk to align the model's predictions on both source and target domains. Although the upper bounds on the target risk (Ben-David et al., 2010; Zhao et al., 2019) contain an *incomputable* term (which depends on inaccessible target labels) that can make adaptation arbitrarily difficult, in practice, optimizing the computable terms in these bounds often appears to be the bottleneck (Ganin et al., 2016; Damodaran et al., 2018; Nguyen et al., 2022a). In this paper, we study this phenomenon in detail using optimal transport (OT) (Peyré & Cuturi, 2019) and explore *gradual shift* (GS) and *probabilistic margin* (PM) assumptions to circumvent the *entanglement* term that appears in the bounds based on OT (Koç et al., 2025). We show that under these assumptions, effective minimization of the computable part of these OT bounds — called the Wasserstein Regularized Risk (WRR) — will lead to the minimization of target risk.

Unfortunately, the addition of a domain alignment regularizer makes the optimization problem more difficult than a purely source risk based ERM objective. Indeed, even with linear models, the augmented objective function becomes non-convex, see for instance Appendix B. This increased difficulty in optimization partially explains the observation that the performance of domain matching methods are often not robust to even small changes in hyperparameters, model architecture, or distribution shift (Gul-

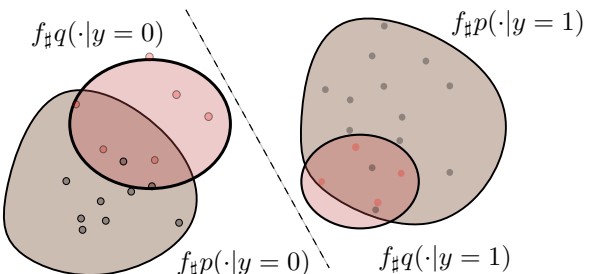

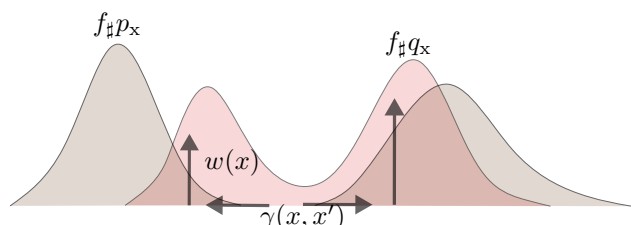

*(a)* A case where there is significant overlap between the source and target output distributions. In such cases, domain matching by trying to align the two distributions exactly can constrain the model capacity unnecessarily (and diminish its discriminative power).

*(b)* The *tightening* of the domain adaptation bounds by weighting the source (output) distribution can equivalently be seen as a *vertical* alignment, apart from the *horizontal* alignment of pure domain matching.

*Figure 1.* During unsupervised domain adaptation (UDA), if the source and target feature densities (where the alignment takes place) share some support, then the UDA bounds can be tightened by looking for the best weighted source distribution to compare against. This implies that the model capacity need not be unnecessarily constrained to match the two domains exactly: only the out-of-support part of the target distribution need to be aligned with the closest weighted source distribution. We use this idea to propose a more relaxed optimal transport (OT) based alignment strategy in this paper and highlight the link to *unbalanced* OT. We also explore assumptions under which such alignment is guaranteed to produce low target risk classifiers.

rajani & Lopez-Paz, 2021; Koç et al., 2025). As a partial remedy, we propose introducing *importance weighting* to the WRR objective, dubbed *weighted* WRR ($W^2R^2$). This leads to a relaxation where OT distances are replaced with *semi-relaxed* OT (Le et al., 2021; Fukunaga & Kasai, 2022), which, whenever entropy regularization is considered, has a closed form softmax solution. Without entropy regularization, we show that the optimization is identical to a ($k = 1$) *nearest neighbor* based alignment strategy, which potentially has high variance (Mack, 1981).

In practice, to avoid collapsing to a nearest neighbor solution, we regularize the semi-relaxed case, obtaining an *unbalanced* OT (UOT) problem — a variant of OT where mass is not preserved (Liero et al., 2018; Chizat et al., 2018). We show that using UOT in $W^2R^2$ leads to the state-of-the-art results in a variety of standard DA benchmarks. More importantly, $W^2R^2$ performs robustly over various distribution shift scenarios under different models without any hyperparameter tuning between experiments. Moreover, these results suggest that most of the standard deep learning models and datasets used in DA experiments respect the introduced GS and PM assumptions.

**Notation.** We define $[n] \doteq \{1, \ldots, n\}$. We denote the space of inputs $\mathcal{X}$ and labels $\mathcal{Y}$, where $\mathcal{Y}$ is a discrete set $|\mathcal{Y}| < \infty$. For a sample space $\mathcal{Z}$, we denote the simplex on $\mathcal{Z}$ by $\triangle(\mathcal{Z})$. We reserve the symbols $p \in \triangle(\mathcal{X} \times \mathcal{Y})$ and $q \in \triangle(\mathcal{X} \times \mathcal{Y})$ to denote source and target joint distributions, respectively. Their corresponding conditionals are denoted by appropriate subscripts, e.g., $p = p_\mathrm{y} p_{\mathrm{x}|\mathrm{y}}$ with $p_\mathrm{y} \in \triangle(\mathcal{Y})$ and $p_{\mathrm{x}|\mathrm{y}}(\cdot \mid y) \in \triangle(\mathcal{X})$ for all $y \in \mathcal{Y}$. For a measure $\mu \in \triangle(\mathcal{Z}_1)$, the pushforward measure

w.r.t. the measurable function $h \colon \mathcal{Z}_1 \to \mathcal{Z}_2$ is defined as $(h_\sharp \mu)(U) = \mu(h^{-1}(U))$ for any measurable subset $U \in \mathcal{Z}_2$. The support of a measure $\mu$ is denoted by $\mathrm{supp}(\mu)$, and the support of a conditional measure's pushforward by $\mathrm{supp}_f(\mu_{\mathrm{x}|\mathrm{y}} \mid y) \doteq \mathrm{supp}((f_\sharp \mu_{\mathrm{x}|\mathrm{y}})(\cdot \mid y))$. The source risk of a classifier $f \colon \mathcal{X} \to \triangle(\mathcal{Y})$ w.r.t. a loss function $\ell$ is denoted by $\epsilon_p(f) = \mathbb{E}_{(x,y) \sim p}[\ell(f(x), y)]$, with the corresponding definition for $\epsilon_q(f)$. With some abuse of notation, we may implicitly consider elements of $y \in \mathcal{Y}$ as elements of $\triangle(\mathcal{Y})$, i.e., by taking $y$'s one-hot vector. To this end, we will consider loss functions $\ell \colon \triangle(\mathcal{Y}) \times \triangle(\mathcal{Y}) \to \mathbb{R}$. We define the Iverson brackets $[\![\texttt{pred}]\!]$ (Knuth, 1992), which evaluate to 1 when $\texttt{pred}$ is true and 0 otherwise.

## 2. Background and Related Work

Throughout the rest of the paper, when we write domain adaptation (DA), we invariably refer to the *unsupervised* case, where labels from the target domain are completely absent during training.

### 2.1. Optimal Transport

DA is concerned with the discrepancies of distributions and in this work we will utilize Optimal Transport (OT) to quantify the distance between them.

**Definition 2.1** (Wasserstein distance). Let $c \colon \mathcal{X} \times \mathcal{Y} \to \mathbb{R}_+$ be a (non-negative) cost function. The $\alpha$-Wasserstein distance w.r.t. $c$ between measures $\mu \in \triangle(\mathcal{X})$ and $\nu \in \triangle(\mathcal{Y})$ is given by

$$W_{\alpha,c}(\mu, \nu) \doteq \left( \inf_{\gamma \in \Gamma(\mu, \nu)} \int_{\mathcal{X} \times \mathcal{Y}} c(x, y)^\alpha \, \mathrm{d}\gamma(x, y) \right)^{1/\alpha},$$

where $\Gamma(\mu, \nu) \doteq \left\{ \gamma \in \triangle(\mathcal{X} \times \mathcal{Y}) : \pi_{1\sharp}\gamma = \mu, \pi_{2\sharp}\gamma = \nu \right\}$ and $\pi_1(x,y) = x$, $\pi_2(x,y) = y$ are projection functions for the corresponding marginal constraints.

Whenever the cost function $c$ is a metric, the $\alpha$-Wasserstein distance can be shown to be a metric between probability distributions that have finite moments of order up to $\alpha$ (Villani et al., 2009). In the discrete case, Wasserstein distances and the underlying OT problem can be computed using linear programming, while entropy-regularized variants can be solved using Sinkhorn iterations (Peyré & Cuturi, 2019).

## 2.2. Domain Adaptation Upper Bounds

We first introduce the notion of entanglement (Koç et al., 2025) that plays an important role in OT based DA analysis.

**Definition 2.2.** For a model $f : \mathcal{X} \to \triangle(\mathcal{Y})$, the label entanglement $\mathcal{E}_{\mathrm{y}}(f)$ and the prediction entanglement $\mathcal{E}_{\hat{\mathrm{y}}}(f)$ are defined as

$$\mathcal{E}_{\mathrm{y}}(f) \doteq \int_{\triangle(\mathcal{Y}) \times \triangle(\mathcal{Y})} W_{1,\ell}(p_{\mathrm{y|f}}(\cdot \mid \hat{y}), q_{\mathrm{y|f}}(\cdot \mid \hat{y}')) \, \mathrm{d}\gamma_{\mathrm{f}}^{\star}(\hat{y}, \hat{y}'),$$

$$\mathcal{E}_{\hat{\mathrm{y}}}(f) \doteq \int_{\mathcal{Y} \times \mathcal{Y}} W_{1,\ell}(p_{\mathrm{f|y}}(\cdot \mid y), q_{\mathrm{f|y}}(\cdot \mid y')) \, \mathrm{d}\gamma_{\mathrm{y}}^{\star}(y, y'),$$

where $p_{\mathrm{y|f}}(y \mid \hat{y})$ and $q_{\mathrm{y|f}}(y \mid \hat{y}')$ correspond to the densities $p, q$ conditioned on $\hat{y} = f(x)$ and $\hat{y}' = f(x')$, while $p_{\mathrm{f|y}}$ and $q_{\mathrm{f|y}}$ denote the pushforwards of $p_{\mathrm{x|y}}$ and $q_{\mathrm{x|y}}$ conditioned on $y$ and $y'$, respectively: $f_{\sharp}p_{\mathrm{x|y}}, f_{\sharp}q_{\mathrm{x|y}}$. The couplings $\gamma_{\mathrm{f}}^{\star}$ and $\gamma_y^{\star}$ are OT plans between pairs $f_{\sharp}p_{\mathrm{x}}, f_{\sharp}q_{\mathrm{x}}$ and $p_y, q_y$.

The two entanglement terms defined above are examples of conditional discrepancies that appear in DA bounds alongside marginal distances (Zhao et al., 2019; Wu et al., 2019). We now introduce an upper bound that follows from the change of measure inequalities.

**Lemma 2.1** (Koç et al. (2025, Corollary 3.1)). *Suppose the loss function $\ell : \triangle(\mathcal{Y}) \times \triangle(\mathcal{Y}) \to \mathbb{R}$ is a metric and that the classifier $f : \mathcal{X} \to \triangle(\mathcal{Y})$ is surjective, then*

$$\epsilon_q(f) \leq \epsilon_p(f) + W_{1,\ell}(f_{\sharp}p_{\mathrm{x}}, f_{\sharp}q_{\mathrm{x}}) + \mathcal{E}_{\mathrm{y}}(f), \quad (1)$$

$$\epsilon_q(f) \leq \epsilon_p(f) + W_{1,\ell}(p_y, q_y) + \mathcal{E}_{\hat{\mathrm{y}}}(f). \quad (2)$$

We remind the reader that a loss function $\ell$ is a metric whenever it is symmetric, non-negative, and respects the triangle inequality. In addition, the identity of indiscernibles needs to hold, i.e., $\ell(y, y') = 0 \iff y = y'$. Although DA bounds such as the one above often require triangle inequality to hold, semimetrics can also be used as loss functions by relaxing the triangle inequality to an approximate version (Crammer et al., 2008; Wang & Mao, 2023), see for instance Koç et al. (2025, Appendix D). The assumption on the surjectivity of $f$ can be relaxed to hold for a suitable subset of $\triangle(\mathcal{Y})$.

**Wasserstein Regularized Risk.** Without target labels, the entanglement $\mathcal{E}_{\mathrm{y}}(f)$ is not computable. Hence, we define the *Wasserstein Regularized Risk* (WRR) objective as the computable part of the bound in (1): $\mathcal{R}_{\lambda}(f) \doteq \epsilon_p(f) + \lambda W_{1,\ell}(f_{\sharp}p_{\mathrm{x}}, f_{\sharp}q_{\mathrm{x}})$, where $\lambda > 0$ is the strength of the DA regularization.

## 2.3. Related Work

Our work has close connections to various ideas in the literature, which we briefly mention below and discuss in more detail in Appendix C.

DA upper bounds first appeared in the seminal work of Ben-David et al. (2010), where the authors introduced the $H\triangle H$-divergence to quantify the distance between the marginal distributions relevant to DA. These bounds were extended to the family of $f$-divergences (Acuna et al., 2021) and to Integral Probability Metrics (IPMs), including OT (Shen et al., 2018; Courty et al., 2017) and Maximum Mean Discrepancy (MMD) based bounds (Long et al., 2015). OT-based bounds were recently refined in Koç et al. (2025).

Including importance weights in domain matching (implicitly or explicitly) via modified divergences has been explored previously, see, e.g., Wu et al. (2019); Tachet des Combes et al. (2020); Johansson et al. (2022). Unbalanced OT (UOT) has also been considered as a heuristic solver that can align distributions in a more outlier-robust manner (Fatras et al., 2021; Nguyen et al., 2022b) compared to OT.

**Assumptions.** Our Gradual Shift (GS) assumption is related to the *connectedness* assumption of Wu et al. (2019), who used it similarly to limit the effect of the conditional discrepancy terms in the DA bounds. Note that the source and target conditional supports were assumed in Wu et al. (2019) to be connected in the input space, which is often an unrealistic assumption, e.g., in the case of images (where arbitrary translations of objects in the scene occur independently of the latent features). The GS assumption in our work instead: (i) relaxes such connectedness to a (invariant) lower-dimensional latent space where the class-conditionals are *ordered* w.r.t. their distances from the source distribution; and (ii) captures the empirical observation that deep neural networks (DNNs), although notoriously good at picking up spurious correlations in the data (Geirhos et al., 2020), have to resort to some invariant *latent* features to separate the source class conditionals accurately. It is this latent space, captured at least partially by DNNs that can separate source with large margin, that is assumed to be shifting gradually in our case (see Theorem 3.1). In Appendix E we relate our GS and PM assumptions further to probabilistic Lipschitzness and anti-collapse assumptions often considered in the literature (Bubeck & Sel-

lke, 2021; Wang et al., 2024).

In a sense our paper can be considered as an extension of the above mentioned key papers by Wu et al. (2019); Koç et al. (2025); Fatras et al. (2021), with the contribution centering around our use of importance weighting to relax the OT-based domain alignment strategy and derive the particular UOT formulation.

## 3. Assumptions for Successful Domain Alignment

The WRR objective $\mathcal{R}_\lambda$ introduced above appears (with $\lambda = 1$) in the upper bound (1) of Lemma 2.1. As it does not depend on target labels, it can be minimized as part of a domain alignment strategy (Damodaran et al., 2018; Courty et al., 2017; Koç et al., 2025). However, an obvious limitation of such an approach is that the label entanglement $\mathcal{E}_y(f)$ can be arbitrary large even when the WRR is low (Koç et al., 2025).

In the following section, we introduce two assumptions on the distribution shift and the hypothesis class in order to control and tame this incomputable term during WRR optimization. This leads us to our central theoretical result (Theorem 3.1) which bounds the target risk by a multiple of $\mathcal{R}_\lambda$ only. Interestingly, our result motivates and grounds the hyperparameter selection of $\lambda$ in $\mathcal{R}_\lambda$ based on the introduced assumptions.

To motivate the need for further assumptions, we start by introducing a lower bound that complements Lemma 2.1's upper bound.[1]

**Lemma 3.1** (Joint error lower bounds, Koç et al. (2025, Lemma B.1)). *Suppose the loss function $\ell\colon \triangle(\mathcal{Y}) \times \triangle(\mathcal{Y}) \to \mathbb{R}$ is a metric and that the classifier $f\colon \mathcal{X} \to \triangle(\mathcal{Y})$ is surjective, then for all $f \in \mathcal{H}$*

$$W_{1,\ell}(p_y, q_y) \leq \mathcal{E}_y(f) + W_{1,\ell}(f_\sharp p_x, f_\sharp q_x)$$
$$\leq \epsilon_p(f) + \epsilon_q(f) + 2W_{1,\ell}(f_\sharp p_x, f_\sharp q_x),$$

*which when combined with Lemma 2.1 yields*

$$W_{1,\ell}(p_y, q_y) - \mathcal{R}_2(f) \leq \mathcal{E}_y(f) - \mathcal{R}_1(f)$$
$$\leq \epsilon_q(f)$$
$$\leq \mathcal{E}_y(f) + \mathcal{R}_1(f).$$

The above lower bound can be seen as an OT variant of the lower bounds reported in the literature (Zhao et al., 2019; Wu et al., 2019). Lemma 3.1 implies that minimizing WRR can actually *inflate* the target error if the label entanglement $\mathcal{E}_y(f)$ of the optimized hypothesis $f$ is non-negligible. As

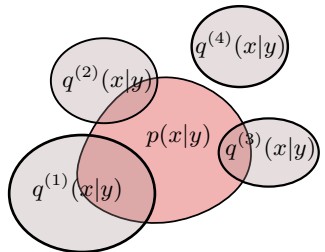

*Figure 2.* An example illustrating *gradual* shift in the target distribution $q$. The gradual shift assumption, together with the probabilistic margin assumption, is utilized in this work to control the growth of conditional discrepancy terms (such as the OT-based entanglement) that require target labels in DA upper bounds. These assumptions help to explain the success of domain alignment strategies that are effective at optimizing their *unsupervised* objectives without requiring target labels.

a result, we need to explore various assumptions (on the distribution shifts and the model) that allow low target error to be achievable for approaches that minimize WRR.

We first introduce our GS assumption, that is a variant of assumptions previously explored in the literature (Koç et al., 2025; Wu et al., 2019).

**Assumption 3.1** (Gradual Shift Assumption). The GS assumption $\mathrm{GS}(a, b, s, \varepsilon)$ holds for joint densities $p$ and $q$ and a classifier $f$ if there exists $q_{x|y}^{(i)}(\cdot \mid y) \in \triangle(\mathcal{X})$ and $a \leq r_i \leq b$ such that for all $y \in \mathcal{Y}$

$$q_{x|y}(x \mid y) = \sum_{i=1}^{s} r_i q_{x|y}^{(i)}(x \mid y),$$

where $(i-1)\varepsilon \leq \ell(\hat{y}, \hat{y}') \leq i\varepsilon$ for $\hat{y} \sim (f_\sharp p_{x|y})(\cdot \mid y)$, $\hat{y}' \sim (f_\sharp q_{x|y}^{(i)})(\cdot \mid y)$ almost surely.

See Figure 2 for an illustration of the assumption. Note that the shifting distributions do not have to be *connected* (Wu et al., 2019) to each other: it is only enough to be able to sort them w.r.t. their distances (with small enough $\varepsilon$ from the source distribution). Secondly, unlike the usual gradual shift assumption employed in, e.g., Kumar et al. (2020), the indices $i$ of the shifting densities $q_{x|y}^{(i)}$ are not assumed to be available, and instead the shifting data are mixed into a single distribution $q$ during DA.

The GS assumption is very flexible due to rich parameterization and any pair of densities $p, q$ can satisfy it for some set of parameters $a, b, s, \varepsilon$: the parameters $a$ and $b$ in effect control the *sampling* (or equivalently, observability) of the shift and $s$ captures the notion of shift *duration* in a discrete setting.[2] Finally, the parameter $\varepsilon$ governs the shift

---

[1]This result was not explicitly stated in Koç et al. (2025), but was used to relate the two entanglement terms in Definition 2.2.

[2]It is possible to extend the definition into a flow formulation, by letting the four parameters depend on two continuous but time-dependent flow and observability parameters instead.

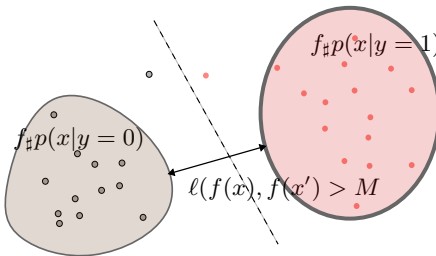

*Figure 3.* An example illustrating the *probabilistic margin* (PM) assumption. If a source feature distribution $f_\sharp p$ satisfies the $\mathrm{PM}(M, \eta)$ assumption with high enough probability $1 - \eta$ for some $M > 0$, then most samples drawn from a conditional will be separated with margin at least $M$ from the support of the other conditionals.

intensity, and we will show that bounding this parameter has implications for DA.

Our second assumption guarantees a certain degree of separability within the source distribution.

**Assumption 3.2** (Probabilistic Margin Assumption)**.** The Probabilistic Margin assumption $\mathrm{PM}(M, \eta)$ holds for a joint distribution $p$ and a classifier $f$ if for all $y \neq y'$,

$$\Pr_{\hat{y} \sim (f_\sharp p_{\mathrm{x}|\mathrm{y}})(\cdot|y)} \left( \inf_{\hat{y}' \in \mathrm{supp}_f(p_{\mathrm{x}|\mathrm{y}}|y')} \ell(\hat{y}, \hat{y}') \geq M \right) \geq 1 - \eta.$$

This assumption is also similar in flavor to assumptions previously explored in Wu et al. (2019). See Figure 3 for an illustration.

**Analysis.** We now use the GS and PM assumptions to present a upper bound on the target risk that does not depend on the label entanglement term of Lemma 2.1. Note that our bound can be solely expressed as a scalar multiple of $\mathcal{R}_\lambda(f)$ for a $\lambda > 0$, which helps explain the good performance observed in some experiments where WRR is the sole quantity being minimized (Courty et al., 2017; Damodaran et al., 2018; Koç et al., 2025). An important quantity in these bounds is the *maximum label ratio* $\rho \doteq \max_{y \in \mathcal{Y}} q_{\mathrm{y}}(y)/p_{\mathrm{y}}(y)$.

**Theorem 3.1.** *Assume* $\mathrm{GS}(a, b, s, \varepsilon)$ *and* $\mathrm{PM}(M, \eta)$ *hold for densities* $p, q$ *and a surjective classifier* $f$*; and loss* $\ell$ *is a metric. Then if* $1 - |\mathcal{Y}|\eta > 0$ *and there exists a* $d \in \mathbb{Z}_+$ *such that* $\varepsilon \leq \frac{M}{2d}$*, then*

$$\epsilon_q(f) \leq \rho \mathcal{R}_\lambda(f), \tag{3}$$

*where* $\lambda = \frac{s(s+1)}{d(d-1)} \frac{b}{\rho a (1 - |\mathcal{Y}|\eta)}$*.*

*Proof Sketch.* Intuitively, the result follows from the fact that the GS and PM assumptions force the shifting conditionals with significant mass (at least $a$ for each $i \in [s]$) to inflate the marginal distance for large enough margin.

See Figure 4 for an illustration. Bounding the conditional Wasserstein distances (which require target labels to evaluate) using the GS parameter $\varepsilon$, we then show, using a worst-case analysis, that these conditional distances can be bounded from above by a constant multiple of the marginal distance. Note that $d$ appearing in the bound can be set to $s$, whenever the marginal distances are low enough, whose precise formulation we leave to the appendix (see Corollary A.2). □

Theorem 3.1 uses a worst-case analysis to show that under the stated assumptions the target risk can be kept small whenever the WRR value is small as well.[3] The theorem can be used to justify the empirical observation that DA algorithms (often minimizing the source risk together with a marginal distance) can *work well* for small enough (conditional and label) shifts and extends it to the settings where the conditional shifts are not necessarily small (but their *increments* $\varepsilon$ are).

On the other hand, the theorem also suggests that during the WRR optimization, it may not be wise to just set $\lambda = 1$. A naive application of Lemma 2.1 where the entire label entanglement term would be ignored, would be equivalent to setting $\lambda = 1$. However, as per Theorem 3.1, under reasonable distribution shift assumptions, we should instead adapt $\lambda$ according to the difficulty of the problem. For instance, the higher the probability that the margin holds (larger $1 - |\mathcal{Y}|\eta$) and the smoother the shifting distributions are (e.g., $b$ and $a$ are close to each other), the smaller the $\lambda$ we require for the upper bound to hold. In practice, this fits with our intuition that for less connected distribution shifts, a stronger regularization (high $\lambda$) can be beneficial.

Note that when minimizing $\mathcal{R}(f)$ during domain matching, the assumptions on the GS and PM parameters need to hold uniformly for some class $f \in \mathcal{F} \subset \mathcal{H}$. For instance, during gradient descent updates on the neural network parameters $\theta_k$, the sequence of models $f_k$ need to have enough regularity such that they do not *hide* the shift at some intermediate $k$, effectively blowing up the $\lambda$ term in the bound (via $b/a \to \infty$). This can happen if, e.g., the optimized network resorts to using *spurious* features exclusively.

## 4. Weighted Wasserstein Regularized Risk

Although Theorem 3.1 suggests that minimizing WRR $\mathcal{R}_\lambda(f)$ for some sufficiently large $\lambda$ will decrease the target risk $\epsilon_q(f)$ under the GS and PM assumptions, in practice, optimizing WRR can be difficult. Indeed, optimization difficulties arise even for linear models, as we show in the Appendix B. We propose a weighted variant of WRR to

---

[3]Note that the bound of Koç et al. (2025, Theorem 4.1) cannot be used to control the target risk for GSs with large $s$.

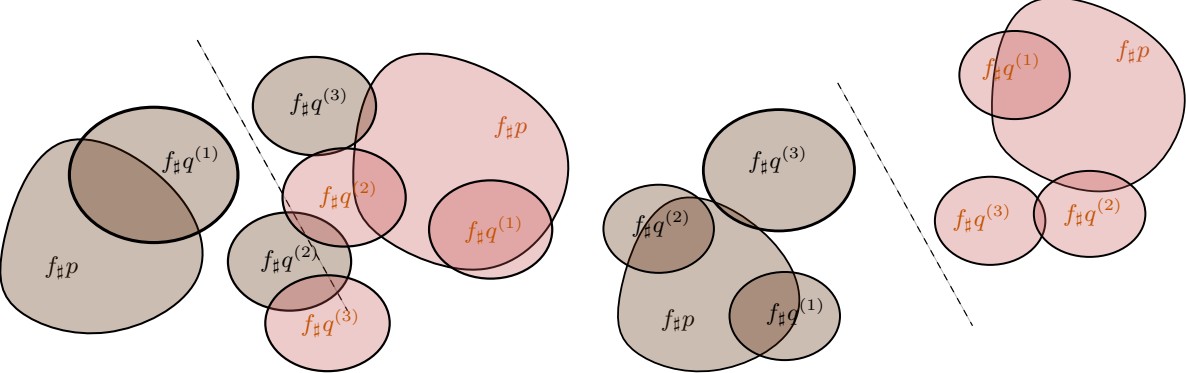

*(a)* Given enough gradual shift, the accuracy of a good source classifier even with large margin can drop significantly. The ▮ brown distributions are conditioned by $y = 0$ and the ▮ red distributions are conditioned by $y = 1$.

*(b)* If the domain matching performance is good (i.e., Wasserstein regularized risk is low), then necessarily the conditionals will be aligned (i.e., entanglement will be low) and the target accuracy will be high. Color legend identical to (a).

*Figure 4.* Given the probabilistic margin $\mathrm{PM}(M, \eta)$ of the classifier and gradual shift $\mathrm{GS}(a, b, s, \varepsilon)$ in the feature space, we show that low source risk and low marginal Wasserstein distance implies low target risk. This is because, the margin assumption forces any shifting conditionals with significant mass to inflate the marginal distance for small enough $\varepsilon < M/2$. Note that the shifting conditionals do not need to be *connected* to each other: the assumption is satisfied whenever the distances to the source conditional are ordered by $\varepsilon$.

alleviate these optimization difficulties, where we are interested to find a weighted source distribution that is easier to align with the target distribution when optimized.

First note that the upper bound on the target risk in Lemma 2.1 holds for any source distribution $p$. As such, instead of directly using the source distribution $p$, we seek to learn *importance weights* (IW) $w \colon \mathcal{X} \to \mathbb{R}_+$ which would tighten the bounds that we are minimizing. We will denote the IW version of the source distribution as $p^w(x, y) = w(x)p(x, y)$, where we will assume that $w$ is appropriately normalized. Similarly, we define weighted versions of marginals and conditionals using a superscripted $w$. Optimizing IW together with the model parameters yields a joint minimization problem that we dub *weighted WRR* ($\mathrm{W}^2\mathrm{R}^2$):

$$\min_{f, w} \; \mathcal{L}_\lambda(f, w) \doteq \epsilon_p^w(f) + \lambda W_{1,\ell}(f_\sharp p_{\mathrm{x}}^w, f_\sharp q_{\mathrm{x}}), \quad (4)$$

where the weighted source risk is defined as

$$\epsilon_p^w(f) \doteq \mathop{\mathbb{E}}_{(x,y) \sim p} \left[ w(x)\ell(f(x), y) \right].$$

Tightening the upper bound (3) by looking for the best weighted source distribution can be equivalently seen as *relaxing* the domain alignment. Examining the second term in (4), instead of trying to align the two distributions $f_\sharp p_{\mathrm{x}}$ and $f_\sharp q_{\mathrm{x}}$ completely via model parameter optimization, the problem is relaxed by reweighting the source distribution such that its pushforward $f_\sharp p_{\mathrm{x}}^w$ is easier to align. Effectively, this relaxed alignment ignores the mass with shared support as long as the source risk is low on that mass. See Figure 1 for a visual illustration.

Yet another perspective is provided through the lens of *model capacity*. By relaxing the domain matching to the *out-of-source-support* case, one can further prevent the alignment from constraining the model capacity unnecessarily.[4] The source risk of such a weighted alignment procedure can achieve lower values, reciprocally allowing the model to guarantee larger margins and strengthening GS.

**Analysis.** The target risk bound (4) presented in Theorem 3.1 can be straightforwardly extended to the weighted setting, as long as the GS and PM assumptions *also depend on* the IW $w$. We will need the weighted variant of the previously introduced maximum label ratio $\rho^w \doteq \max_{y \in \mathcal{Y}} q_{\mathrm{y}}(y)/p_{\mathrm{y}}^w(y)$.

**Corollary 4.1.** *Assume* $\mathrm{GS}(a, b, s, \varepsilon^w)$ *and* $\mathrm{PM}(M^w, \eta^w)$ *hold for distributions* $p^w$, $q$, *a surjective classifier* $f$ *and normalized importance weights* $w$*; loss* $\ell$ *is a metric. If* $\varepsilon^w \leq M^w/(2d)$ *for some* $d \in \mathbb{Z}_+$*, we have that*

$$W_{1,\ell}(p_{\mathrm{y}}^w, q_{\mathrm{y}}) - \mathcal{L}_2(f, w) \leq \epsilon_q(f) \leq \rho^w \mathcal{L}_{\lambda^w}(f, w), \quad (5)$$

*for* $\lambda^w = \frac{s(s+1)}{d(d-1)} \frac{b}{a(1-|\mathcal{Y}|\eta^w)\rho^w}$.

For simplicity of discussion, we assume that $a$, $b$, $s$, and $d$ are fixed, but that GS and PM hold for different parameters $\varepsilon^w$, $M^w$, and $\eta^w$ depending on our choice of normalized $w$. As a result, there is a potential tradeoff between the ease of optimization and the values of the upper and lower bounds:

---

[4]In general, model capacity is *constrained* whenever the WRR objective is lower bounded by some term, while the ERM risk is not. We present such a lower bound for the linear regression case in Appendix B.

optimizing for $w$ in (4) may actually *increase* the target risk by pushing up the label shift term $W_{1,\ell}(p_{\mathrm{y}}^w, q_{\mathrm{y}})$ in the lower bound, or may break up the GS structure needed to guarantee low target risk (e.g., by increasing $\varepsilon$ inadvertently between the weighted source and the shifting target distribution). Of course, in the most advantageous case, we may find weights which both bring the label distributions closer and strengthen the GS link between the distributions, enabling an easier alignment that also lowers the target risk.

Next, we show that the inner optimization of (4) w.r.t. $w$ corresponds to a semi-relaxed OT problem. As the discussion above suggests that a poor choice in weights can make the bound looser (despite any optimization convenience), we further consider regularize the semi-relaxed problem.

A convenient way of expressing this regularization is through *unbalanced optimal transport* (UOT), which relaxes the marginal constraints of balanced Wasserstein distance (Definition 2.1) into Lagrangian penalties, where the regularizers (depending on the penalty coefficients $\tau_1$ and $\tau_2$) can allow mass to be created, varied, or destroyed (hence the origin of the term *unbalanced*) (Liero et al., 2018; Chizat et al., 2018). In our setting, this is precisely the mechanism that will let us interpolate between the fully semi-relaxed formulation and a more stable optimization problem.

**Definition 4.1** (Unbalanced Optimal Transport)**.** The Unbalanced Optimal Transport (UOT) between measures $\mu \in \triangle(\mathcal{X})$ and $\nu \in \triangle(\mathcal{Y})$ is given by

$$\mathrm{UOT}(\mu, \nu) \doteq \inf_{\gamma \in \triangle(\mathcal{X} \times \mathcal{Y})} \int_{\mathcal{X} \times \mathcal{Y}} c(x, y) \, \mathrm{d}\gamma(x, y)$$
$$+ \tau_1 \mathrm{KL}(\pi_{1\sharp}\gamma \parallel \mu) + \tau_2 \mathrm{KL}(\pi_{2\sharp}\gamma \parallel \nu).$$

In the limit as $\tau_1 \to 0$ and $\tau_2 \to \infty$ (or vice versa), we get a *semi-relaxed* optimal transport problem, where one of the OT constraints drops entirely. Note that UOT can be defined in general using any divergence function in the penalty terms (Peyré & Cuturi, 2019), but we choose to use the canonical Kullback-Leibler (KL) divergence (Kullback & Leibler, 1951) for simplicity.

**Connection to Semi-Relaxed OT.** An appealing feature of $\mathrm{W}^2\mathrm{R}^2$ is that the inner optimization over the importance weights can be rewritten in a form that exposes a direct connection to semi-relaxed OT. Expanding $W_{1,\ell}(f_\sharp p_{\mathrm{x}}^w, f_\sharp q_{\mathrm{x}})$ and using the marginal constraint $\pi_{1\sharp}\gamma_{\mathrm{f}}^w = f_\sharp p_{\mathrm{x}}^w$, the inner problem becomes

$$\inf_{w,\gamma_{\mathrm{f}}^w} \int_{\mathcal{Y}^3} \ell(\hat{y}, y) + \lambda\ell(\hat{y}, \hat{y}') \, \mathrm{d}\gamma_{\mathrm{f}}^w(\hat{y}, \hat{y}') \, \mathrm{d}p_{\mathrm{y}|\mathrm{f}}(y \mid \hat{y}),$$
$$s.t. \ \pi_{1\sharp}\gamma_{\mathrm{f}}^w = f_\sharp p_{\mathrm{x}}^w \quad \text{and} \quad \pi_{2\sharp}\gamma_{\mathrm{f}}^w = f_\sharp q_{\mathrm{x}}, \quad (6)$$

where the minimization over $w$ is taken outside the minimization over the coupling $\gamma_{\mathrm{f}}^w$ (hence the superscript $w$).[5] This reformulation will let us eliminate the dependence on $w$ entirely.

For simplicity, we consider the discrete version of the optimization problem. Suppose we have $m$ samples from the source distribution $\{(x_i, y_i) \sim p\}_{i=1}^m$ and $n$ samples from the target marginal distribution $\{x_j' \sim q_{\mathrm{x}}\}_{j=1}^n$. Then (6) can be written as

$$\min_{\Gamma \in \mathbb{R}_+^{m \times n}} \min_{w \in \mathbb{R}_+^n} \quad \ell^{\mathrm{T}}w + \lambda\mathrm{tr}(C\Gamma^{\mathrm{T}}),$$
$$s.t. \quad \Gamma\mathbb{1}_n = w \quad \text{and} \quad \Gamma^{\mathrm{T}}\mathbb{1}_m = \tfrac{1}{n}\mathbb{1}_n, \quad (7)$$

where $\mathbb{1}.$ denotes the vector of ones and the cost matrix $C \in \mathbb{R}^{m \times n}$ consists of elements $C_{ij} = \ell(f(x_i), f(x_j'))$. We also write $\ell_i = \ell(f(x_i), y_i)$ for the source losses. As the first marginal constraint already determines the optimized weights, we can plug-in $w = \Gamma\mathbb{1}_n$ into the objective:

$$\mathcal{L}_\lambda(f) = \min_{\Gamma \in \mathbb{R}_+^{m \times n}} \mathrm{tr}(\tilde{C}^\lambda\Gamma^{\mathrm{T}}), \quad s.t. \ \Gamma^{\mathrm{T}}\mathbb{1}_m = \tfrac{1}{n}\mathbb{1}_n, \quad (8)$$

where $\tilde{C}^\lambda \doteq \ell\mathbb{1}_n^{\mathrm{T}} + \lambda C$. Note that the resulting optimization problem (8) is a linear program that does not include the weights $w$. This corresponds to a semi-relaxed OT problem with a single marginal constraint (Le et al., 2021; Fukunaga & Kasai, 2022).[6]

**Nearest Neighbors.** As there is only one marginal constraint in (8), one can actually solve the optimal coupling and the corresponding weights (noting $w_{\mathrm{nn}}^\star = \Gamma_{\mathrm{nn}}^\star\mathbb{1}_n$) in closed form. Indeed, the optimization problem is solved by assigning all the mass into nearest neighbors w.r.t. the cost matrix $\tilde{C}^\lambda$:

$$(\Gamma_{\mathrm{nn}}^\star)_{ij} = \frac{[\![i = \mathrm{nn}(j)]\!]}{n} \quad \text{and} \quad (w_{\mathrm{nn}}^\star)_i = \sum_{j=1}^n \frac{[\![i = \mathrm{nn}(j)]\!]}{n},$$

where $\mathrm{nn}(j) = \arg\min_{i \in [m]} \tilde{C}_{ij}^\lambda$ corresponds to the nearest neighbor of $j$ according to cost matrix $\tilde{C}^\lambda$. Plugging the weight back into the $\mathrm{W}^2\mathrm{R}^2$ objective (4), we get

$$\min_w \mathcal{L}_\lambda(f, w)$$
$$= \frac{1}{n} \sum_{j=1}^n \min_{i \in [m]} \left\{ \ell(f(x_i), y_i) + \lambda\ell(f(x_i), f(x_j')) \right\}. \quad (9)$$

As one can see from (9), using the nearest neighbor based IW $w_{\mathrm{nn}}^\star$ can cause issues in the discrete/empirical measure case. The alignment would only concentrate on the shortest distance pairings $\ell(f(x_i), f(x_j'))$ and ignore the large majority of other source and target pairings.

---

[5]Note that the pushforward of this constraint can be 'distributed' via $(f_\sharp p_{\mathrm{x}}^w)(\hat{y}) = (f_\sharp w)(\hat{y})(f_\sharp p_{\mathrm{x}})(\hat{y})$.

[6]The reduction from importance weighted regular OT to semi-relaxed OT may be of independent interest outside of DA.

**Regularizing to UOT.** One way of avoiding the importance weights from collapsing to a nearest neighbor solution is to reintroduce the marginal constraint that is not present in (8). Instead of using a "hard constraint", we consider the UOT problem described in Definition 4.1 with the KL regularizers. Considering uniform marginals, the entropy regularized UOT problem yields the following variant of Sinkhorn iterations (Peyré & Cuturi, 2019):

$$
\begin{aligned}
u^{k+1} &= \left( \frac{\mathbb{1}_m}{mKv^k} \right)^{\frac{\tau_1}{\tau_1+\epsilon}}, \\
v^{k+1} &= \left( \frac{\mathbb{1}_n}{nK^{\mathrm{T}}u^{k+1}} \right)^{\frac{\tau_2}{\tau_2+\epsilon}},
\end{aligned}
\tag{10}
$$

where $\Gamma^k \doteq \mathrm{diag}(u^k)K\mathrm{diag}(v^k)$ and $K \doteq \exp\left(-\tilde{C}^\lambda/\epsilon\right)$. As $k \to \infty$, the coupling $\Gamma^k$ converges to the optimal coupling of the entropy regularized UOT problem.

It should be noted that different (limiting) choices of $\tau_1$ and $\tau_2$ correspond to different variants of the OT problem. When we take $\tau_1 \to \infty$ and $\tau_2 \to \infty$, standard (balanced) OT is recovered. In our experiments, we pick values of $\tau_1 = 1$ and $\tau_2 = 100$ to mimic the single constraint of (8).

**Entropy Regularized Semi-Relaxed OT.** If we take $\tau_1 = 0$ and $\tau_2 \to \infty$ in (10), we recover an entropy regularized version of the original $\mathrm{W}^2\mathrm{R}^2$ discrete OT (8):

$$
\min_{\Gamma \in \mathbb{R}_+^{m \times n}} \mathrm{tr}\left(\tilde{C}^\lambda \Gamma^{\mathrm{T}}\right) - \epsilon \mathrm{H}(\Gamma), \; s.t. \; \Gamma^{\mathrm{T}}\mathbb{1}_m = \frac{1}{n}\mathbb{1}_n. \tag{11}
$$

Notably, in this case, our Sinkhorn iterations collapses to a single update

$$
(\Gamma_\varepsilon^\star)_{:,j} = \frac{1}{n}\sigma\left(-\tilde{C}_{:,j}^\lambda/\epsilon\right), \tag{12}
$$

where $\sigma$ denotes the softmax function and the notation $M_{:,j}$ corresponds to extracting out the $j$-th column vector of $M$. The weights of this entropy regularized version of (8) can also be easily solved (again using $w_\epsilon^\star = \Gamma_\epsilon^\star \mathbb{1}_n$)

$$
w_\epsilon^\star = \frac{1}{n}\sum_{j=1}^n \sigma\left(-\tilde{C}_{:,j}^\lambda/\epsilon\right). \tag{13}
$$

The weights correspond to the empirical mean of the soft assignments $\sigma(-\tilde{C}_{:,j}^\lambda/\epsilon)$. In other words, by adding entropy regularization, the optimal weights go from the nearest neighbor "hard labels" to their softmax "soft label" counterparts. We can also further verify that taking $\varepsilon \to 0$ indeed yields the nearest neighbor weights $w_{\mathrm{nn}}^\star$.

## 5. Experiments

We evaluate our proposed method $\mathrm{W}^2\mathrm{R}^2$ in various distribution shift *scenarios*.[7] These include the popular *digits*

---

[7]Code is available at https://github.com/okankoc/incorporating_iw_in_ot_da.

datasets with various permutations: (i) MNIST to USPS, (ii) USPS to MNIST, (iii) SVHN to MNIST, and (iv) MNIST to MNIST-M.

In order to test the robustness of our proposed approach, we evaluate our methods over three different models: a multi layered perceptron (MLP) composed of two hidden layers, a standard convolutional neural network (Conv) without batch normalization and pooling, and an ImageNet-pretrained residual network (ResNet18) that includes standard batch normalization and pooling layers. All model parameters are optimized in PyTorch (Paszke et al., 2019) using the Adam optimizer (Kingma & Ba, 2015) and a learning rate of $10^{-3}$, except when the model used is ResNet18, in which case the learning rate is reduced to $10^{-4}$ due to stability issues. We run experiments for 10 epochs with a batch-size of 64 and repeat five times to generate accuracy curves with standard deviations. The first 5 epochs are used for pretraining the model without domain adaptation. Motivated by the need to satisfy the PM assumption (with large $M$ and small $\eta$), we use a margin multi-classification loss (Crammer & Singer, 2001) in order to encourage clear source class-conditional separation during the optimization process.

We compare our proposed approach $\mathrm{W}^2\mathrm{R}^2$ against standard and promising UDA baselines, which are discussed in detail in Appendix D. Due to space constraints, we present the target accuracies at the tenth epoch in Table 1 and include the accuracy curves in the appendix as well. Besides the UDA methods that do not use target labels, we also include two oracles and a standard ERM optimizer, in order to assess the difficulty of the scenarios and to quantify the positive transfer achieved by the UDA approaches in a more realistic way.

**Implementation.** For the $\mathrm{W}^2\mathrm{R}^2$ optimization, we use the mm-UOT solver (Chapel et al., 2021) that directly solves for the optimal coupling using a majorization-minimization method without entropy-regularization. It is implemented as part of the POT library (Flamary et al., 2021) and we find that it performs more reliably compared to the other Sinkhorn-iteration based UOT methods (which can fail to converge at times) provided by that library. Similarly to the *geomloss* library (Feydy et al., 2019), we speed up the solver by disabling the autograd functionality of PyTorch (Paszke et al., 2019) during the iterations, enabling it only at the end, upon (often) successful convergence. We set $\lambda = 1$ and add *unweighted* source risk to the UOT-regularized $\mathrm{W}^2\mathrm{R}^2$ objective of (8)

$$
\begin{aligned}
\frac{1}{m}\ell^{\mathrm{T}}\mathbb{1}_m + \min_{\Gamma \in \mathbb{R}_+^{m \times n}} \mathrm{tr}\left(\tilde{C}^1 \Gamma^{\mathrm{T}}\right) + \mathrm{KL}\left(\Gamma\mathbb{1}_n \,\|\, \tfrac{\mathbb{1}_m}{m}\right) \\
+ 100\mathrm{KL}\left(\Gamma^{\mathrm{T}}\mathbb{1}_m \,\|\, \tfrac{\mathbb{1}_n}{n}\right).
\end{aligned}
\tag{14}
$$

*Table 1.* Summary of test accuracy comparison of different models and unsupervised DA algorithms. ERM and two supervised (access to target labels) oracles algorithms are presented for baseline comparison. Experiments are run over a variety of DA scenarios. Scenarios are only reported if their oracle (LJE and CC) accuracy is over 80%. Hyperparameters are fixed across different scenarios to create a more difficult but realistic evaluation setting. Accuracy and s.t.d. is reported over five trials. The best performing algorithm is boxed.

| | | Oracles | | Baseline | UDA Algorithms | | | |
|---|---|---|---|---|---|---|---|---|
| Scenario | Model | LJE | CC | ERM | DANN | rev-KL | WRR | $W^2R^2$ |
| MNIST $\rightarrow$ USPS | MLP | $0.95 \pm 0.0020$ | $0.89 \pm 0.0040$ | $0.33 \pm 0.0085$ | $0.55 \pm 0.0540$ | $0.62 \pm 0.0110$ | $0.77 \pm 0.0010$ | $\boxed{0.86 \pm 0.0053}$ |
| | ConvNet | $0.97 \pm 0.0000$ | $0.97 \pm 0.0027$ | $0.85 \pm 0.0017$ | $0.89 \pm 0.0053$ | $0.91 \pm 0.0015$ | $\boxed{0.95 \pm 0.0017}$ | $0.94 \pm 0.0033$ |
| | ResNet | $0.97 \pm 0.0026$ | $0.90 \pm 0.0330$ | $0.88 \pm 0.0062$ | $0.88 \pm 0.0037$ | $0.88 \pm 0.0150$ | $0.88 \pm 0.0370$ | $\boxed{0.95 \pm 0.0015}$ |
| USPS $\rightarrow$ MNIST | MLP | $0.95 \pm 0.0013$ | $0.91 \pm 0.0013$ | $0.30 \pm 0.0018$ | $0.42 \pm 0.0110$ | $0.51 \pm 0.0120$ | $0.57 \pm 0.0072$ | $\boxed{0.62 \pm 0.0140}$ |
| | ConvNet | $0.98 \pm 0.0014$ | $0.97 \pm 0.0031$ | $0.72 \pm 0.0150$ | $0.84 \pm 0.0088$ | $0.85 \pm 0.0019$ | $0.91 \pm 0.0052$ | $\boxed{0.95 \pm 0.0064}$ |
| | ResNet | $0.98 \pm 0.0000$ | $0.95 \pm 0.0025$ | $0.73 \pm 0.0077$ | $0.74 \pm 0.0310$ | $0.67 \pm 0.1100$ | $0.90 \pm 0.0066$ | $\boxed{0.93 \pm 0.0070}$ |
| MNIST $\rightarrow$ MNISTM | ConvNet | $0.93 \pm 0.0024$ | $0.90 \pm 0.0014$ | $0.48 \pm 0.0140$ | $\boxed{0.68 \pm 0.0081}$ | $0.53 \pm 0.0064$ | $0.61 \pm 0.0071$ | $0.49 \pm 0.0130$ |
| | ResNet | $0.89 \pm 0.0017$ | $0.84 \pm 0.0015$ | $0.32 \pm 0.0140$ | $0.42 \pm 0.0640$ | $0.10 \pm 0.0070$ | $0.50 \pm 0.0200$ | $\boxed{0.66 \pm 0.0100}$ |
| SVHN $\rightarrow$ MNIST | ResNet | $0.98 \pm 0.0005$ | $0.96 \pm 0.0046$ | $0.76 \pm 0.0120$ | $0.55 \pm 0.1200$ | $0.57 \pm 0.1100$ | $\boxed{0.78 \pm 0.0160}$ | $0.73 \pm 0.0073$ |

We find this setting to work well over the tested distribution shift scenarios. The addition of the unweighted source risk in (14) can be interpreted as further regularizing or *flattening* the optimized weights. Note that, unlike the majority of OT based domain alignment methods (Damodaran et al., 2018; Shen et al., 2018), we match the distributions in *output* space, i.e., the cost matrix $C_{ij} = \ell(f(x_i), f(x'_j))$ in (14) uses the predicted probabilities $f(x_i), f(x'_j) \in \triangle(\mathcal{Y})$.

**Discussion.** We note again that semi-relaxed OT is a limit case of UOT, where the optimized criterion completely ignores the mass (but not the location) of the source dataset when aligning the two (weighted) distributions. In order to reduce the variance of the optimized weights (and as a consequence, of the coupling as well), in (14) we use UOT with a small penalty for the source ($\tau_1 = 1$) and a large penalty ($\tau_2 = 100$) for the target samples. We find that our results are quite robust to the precise setting of these values, as long as $\tau_1$ is significantly above zero. The semi-discrete entropy-regularized OT, with the explicit softmax solution for the optimal coupling given in (11), is competitive with the mm-UOT solver for the ConvNet model in our experiments, but becomes unstable for the MLP model, which we think results from the higher capacity and higher *non-invertibility* of the MLP models (Gilbert et al., 2017): the optimizer easily finds embeddings where most of the target points are aligned to a few source points during the semi-relaxed alignment, breaking the GS structure.

For most of the scenario/model permutations, our proposed approach shows (and maintains) a robust performance, without any hyperparameter tuning between experiments (except for the learning rate change for the ResNet model). Finally we would like to note that the margin loss that we use in (14) cannot be optimized effectively *without pretraining*. If pretraining (where only the source risk is minimized) is disabled, then $W^2R^2$ often stays at chance levels, unable to optimize the objective. In such cases,

we observed that using the Euclidean distance (unlike the cross-entropy loss) could be a better alternative.

## 6. Conclusion

In this paper we introduced the Gradual Shift (GS) and Probabilistic Margin (PM) assumptions to circumvent entanglement, a conditional discrepancy term in optimal transport based upper bounds that cannot be evaluated without target labels. The resulting theory suggests that whenever the GS and PM assumptions are satisfied, a low Wasserstein regularized risk (WRR) will lead to low target risk as well.

Despite the theory, in practice WRR can be a difficult objective to optimize for and the bound can be loose. As a result, we proposed an weighted version of WRR ($W^2R^2$) that tightens the target risk bounds and can be interpreted as applying importance weighting on the WRR. Our experiments show that $W^2R^2$ can beat state-of-the-art methods without heavy hyperparameter fine-tuning. Moreover, the results suggest that the GS and PM assumptions we explored hold in standard DA datasets.[8]

Finally, while we argued throughout the text that keeping the $\lambda$ scale bounded throughout the WRR optimization is enough to drive the source risk to low values, in practice, estimating the GS shift intensity from (labeled) source and unlabeled target samples and constructing upper bounds on $\lambda$ *during* the UDA optimization would make our proposed solution more cautious and flexible, increasing the robustness to spurious features and to poorly selected hyperparameters. We leave this important research direction for future work.

---

[8]We show in Appendix F diagnostics that suggest that both the GS and PM assumptions are often reasonable assumptions to make for the models and scenarios that we consider in the experiments.

## Impact Statement

This paper presents work whose goal is to advance the field of trustworthy Machine Learning, and in particular improve upon the robustness of domain adaptation algorithms currently studied in research. There are many potential societal consequences of our work, which we would like to briefly highlight below.

Whenever the GS and PM assumptions are violated, the algorithms risk creating a sense of false confidence, as it is possible for target risk to be high even when both the source risk and the Wasserstein distance terms (whether balanced or unbalanced) are small. Such failure can manifest itself directly in deterioration of average accuracy, or more subtly, in the uneven performance across target subpopulations.

The algorithmic choices made in the experimental evaluation section of the article, such as using pretraining and the particular solver for UOT, should also be studied in greater detail, as currently the training that we propose is not particularly robust to variations of these choices.

**Acknowledgments.** Masashi Sugiyama was supported by JST ASPIRE Grant Number JPMJAP25B1.

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

# Appendix

## A. Proofs

We first repeat below the assumptions that were presented already in the main text for the reader's convenience. We then present three lemmas that will facilitate the proof of Theorem 3.1.

The GS Assumption allows us to sort the target conditional distributions into $s$ *ordered* distributions (in terms of their distances to the source distribution $p$) having density at least $a$ and at most $b$.

**Assumption A.1** (Gradual Shift Assumption). The Gradual Shift assumption $GS(a, b, s, \varepsilon)$ holds for joint distributions $p$ and $q$ and a classifier $f$ if there exists $q_{x|y}^{(i)}(\cdot \mid y) \in \triangle(\mathcal{X})$ and $a \leq r_i \leq b$ for all $i \in \{1, \ldots, s\}$ and for all $y \in \mathcal{Y}$ such that

$$q_{x|y}(x \mid y) = \sum_{i=1}^{s} r_i q_{x|y}^{(i)}(x \mid y),$$

where

$$\Pr_{\hat{y}, \hat{y}'} ((i-1)\varepsilon \leq \ell(\hat{y}, \hat{y}') \leq i\varepsilon) = 1,$$

and $\hat{y}, \hat{y}'$ being drawn from the conditionals: $\hat{y} \sim f_\sharp p_{x|y}(\cdot \mid y)$ and $\hat{y}' \sim f_\sharp q_{x|y}^{(i)}(\cdot \mid y)$.

**Assumption A.2** (Probabilistic Margin Assumption). The Probabilistic Margin assumption $PM(M, \eta)$ holds for a joint distribution $p$ and a classifier $f$ if for all $y, y' \in \mathcal{Y}, y \neq y'$,

$$\Pr_{\hat{y} \sim (f_\sharp p_{x|y})(\cdot|y)} \left( \inf_{\hat{y}' \in \text{supp}_f(p_{x|y}|y')} \ell(\hat{y}, \hat{y}') \geq M \right) \geq 1 - \eta.$$

The following lemma results from the fact that the $PM(M, \eta)$ and $GS(a, b, s, \varepsilon)$ assumptions force the marginal distribution $p_x$ and $q_{x|y}^{(i)}$ to have disjoint support with high probability, if the gradual shift increments are small enough.

**Lemma A.1.** *Given two joint distributions $p$ and $q$ and a classifier $f$ such that $p, f$ satisfy the $PM(M, \eta)$ assumption and $p, q, f$ satisfy the $GS(a, b, s, \varepsilon)$ assumption, then if $\varepsilon \leq \frac{M}{2d}$ for some $d > 1$*

$$\Pr_{\hat{y} \sim f_\sharp q_{x|y}^{(i)}(\cdot|y)} \left( \inf_{\hat{y}' \in \text{supp}(f_\sharp p_x)} \ell(\hat{y}, \hat{y}') \geq (i-1)\varepsilon \right) > 1 - |\mathcal{Y}|\eta,$$

*for $i \leq d$.*

*Proof.* We first examine the support $\text{supp}(f_\sharp p_x)$. From the definition of a pushforward measure, we have

$$\begin{aligned}
\hat{y}' \in \text{supp}(f_\sharp p_x) &\iff (f_\sharp p_x)(\hat{y}') > 0 \\
&\iff p_x(\{x \mid f(x) = \hat{y}'\}) > 0 \\
&\iff p_y(y')p_{x|y}(\{x \mid f(x) = \hat{y}'\} \mid y') > 0 \quad \text{for some } y' \in \mathcal{Y}.
\end{aligned}$$

The final line follows from noticing the marginalization of the joint needs to be non-zero.

Now restricting $p_y(y) > 0$ for all $y \in \mathcal{Y}$, we have

$$\begin{aligned}
\hat{y}' \in \text{supp}(f_\sharp p_x) &\iff p_{x|y'}(\{x \mid f(x) = \hat{y}'\} \mid y') > 0 \quad \text{for some } y' \in \text{supp}(p_y) \\
&\iff (f_\sharp p_{x|y})(\hat{y}' \mid y') > 0 \quad \text{for some } y' \in \text{supp}(p_y) \\
&\iff \hat{y}' \in \text{supp}_f(p_{x|y} \mid y') \quad \text{for some } y' \in \text{supp}(p_y).
\end{aligned}$$

Now through a union bound, we simplify

$$
\Pr_{\hat{y} \sim f_{\sharp} q_{\mathsf{x}|\mathsf{y}}^{(i)}(\cdot|y)} \left( \inf_{\hat{y}' \in \mathrm{supp}(f_{\sharp} p_{\mathsf{x}})} \ell(\hat{y}, \hat{y}') < (i-1)\varepsilon \right) = \Pr_{\hat{y} \sim f_{\sharp} q_{\mathsf{x}|\mathsf{y}}^{(i)}(\cdot|y)} \left( \min_{y' \in \mathrm{supp}(p_{\mathsf{y}})} \inf_{\hat{y}' \in \mathrm{supp}_f(p_{\mathsf{x}|\mathsf{y}}|y')} \ell(\hat{y}, \hat{y}') < (i-1)\varepsilon \right)
$$

$$
\leq \Pr_{\hat{y} \sim f_{\sharp} q_{\mathsf{x}|\mathsf{y}}^{(i)}(\cdot|y)} \left( \min_{y' \in \mathcal{Y}} \inf_{\hat{y}' \in \mathrm{supp}_f(p_{\mathsf{x}|\mathsf{y}}|y')} \ell(\hat{y}, \hat{y}') < (i-1)\varepsilon \right)
$$

$$
\leq \sum_{y' \in \mathcal{Y}} \Pr_{\hat{y} \sim f_{\sharp} q_{\mathsf{x}|\mathsf{y}}^{(i)}(\cdot|y)} \left( \inf_{\hat{y}' \in \mathrm{supp}_f(p_{\mathsf{x}|\mathsf{y}}|y')} \ell(\hat{y}, \hat{y}') < (i-1)\varepsilon \right).
$$

We now split the summation into two cases, when $y' = y$ and $y' \neq y$. For the former case, we know that from the GS assumption, we have the distance constraints holding almost surely

$$
\Pr_{\hat{y} \sim f_{\sharp} p_{\mathsf{x}|\mathsf{y}}(\cdot|y), \hat{y}' \sim f_{\sharp} q_{\mathsf{x}|\mathsf{y}}^{(i)}(\cdot|y)} (\ell(\hat{y}, \hat{y}') \geq (i-1)\varepsilon) = 1 \implies \Pr_{\hat{y} \sim f_{\sharp} q_{\mathsf{x}|\mathsf{y}}^{(i)}(\cdot|y)} \left( \inf_{\hat{y}' \in \mathrm{supp}_f(p_{\mathsf{x}|\mathsf{y}}|y')} \ell(\hat{y}, \hat{y}') \leq (i-1)\varepsilon \right) = 0.
$$

For the latter case, we consider the following chain of inequalities using the GS assumption as well:

$$
\Pr_{\hat{y} \sim f_{\sharp} q_{\mathsf{x}|\mathsf{y}}^{(i)}(\cdot|y)} \left( \inf_{\hat{y}' \in \mathrm{supp}_f(p_{\mathsf{x}|\mathsf{y}}|y')} \ell(\hat{y}, \hat{y}') < (i-1)\varepsilon \right)
$$

$$
= \int_{\mathcal{Y}} \left[\!\!\left[ \inf_{\hat{y}' \in \mathrm{supp}_f(p_{\mathsf{x}|\mathsf{y}}|y')} \ell(\hat{y}, \hat{y}') < (i-1)\varepsilon \right]\!\!\right] \mathrm{d}f_{\sharp} q_{\mathsf{x}|\mathsf{y}}^{(i)}(\hat{y} \mid y)
$$

$$
\overset{(a)}{\leq} \iint_{\mathcal{Y} \times \mathcal{Y}} \left[\!\!\left[ \inf_{\hat{y}' \in \mathrm{supp}_f(p_{\mathsf{x}|\mathsf{y}}|y')} \ell(\hat{y}'', \hat{y}') - \ell(\hat{y}, \hat{y}'') < (i-1)\varepsilon \right]\!\!\right] \mathrm{d}f_{\sharp} p_{\mathsf{x}|\mathsf{y}}(\hat{y}'' \mid y) \,\mathrm{d}f_{\sharp} q_{\mathsf{x}|\mathsf{y}}^{(i)}(\hat{y} \mid y)
$$

$$
= \iint_{\mathcal{Y} \times \mathcal{Y}} \left[\!\!\left[ \inf_{\hat{y}' \in \mathrm{supp}_f(p_{\mathsf{x}|\mathsf{y}}|y')} \ell(\hat{y}'', \hat{y}') < (i-1)\varepsilon + \ell(\hat{y}, \hat{y}'') \right]\!\!\right] \mathrm{d}f_{\sharp} p_{\mathsf{x}|\mathsf{y}}(\hat{y}'' \mid y) \,\mathrm{d}f_{\sharp} q_{\mathsf{x}|\mathsf{y}}^{(i)}(\hat{y} \mid y)
$$

$$
\overset{(b)}{\leq} \iint_{\mathcal{Y} \times \mathcal{Y}} \left[\!\!\left[ \inf_{\hat{y}' \in \mathrm{supp}_f(p_{\mathsf{x}|\mathsf{y}}|y')} \ell(\hat{y}'', \hat{y}') < (i-1)\varepsilon + i\varepsilon \right]\!\!\right] \mathrm{d}f_{\sharp} p_{\mathsf{x}|\mathsf{y}}(\hat{y}'' \mid y) \,\mathrm{d}f_{\sharp} q_{\mathsf{x}|\mathsf{y}}^{(i)}(\hat{y} \mid y)
$$

$$
= \iint_{\mathcal{Y} \times \mathcal{Y}} \left[\!\!\left[ \inf_{\hat{y}' \in \mathrm{supp}_f(p_{\mathsf{x}|\mathsf{y}}|y')} \ell(\hat{y}'', \hat{y}') < (2i-1)\varepsilon \right]\!\!\right] \mathrm{d}f_{\sharp} p_{\mathsf{x}|\mathsf{y}}(\hat{y}'' \mid y) \,\mathrm{d}f_{\sharp} q_{\mathsf{x}|\mathsf{y}}^{(i)}(\hat{y} \mid y)
$$

$$
= \int_{\mathcal{Y}} \left[\!\!\left[ \inf_{\hat{y}' \in \mathrm{supp}_f(p_{\mathsf{x}|\mathsf{y}}|y')} \ell(\hat{y}'', \hat{y}') < (2i-1)\varepsilon \right]\!\!\right] \mathrm{d}f_{\sharp} p_{\mathsf{x}|\mathsf{y}}(\hat{y}'' \mid y)
$$

$$
= \Pr_{\hat{y} \sim f_{\sharp} p_{\mathsf{x}|\mathsf{y}}(\cdot|y)} \left( \inf_{\hat{y}' \in \mathrm{supp}_f(p_{\mathsf{x}|\mathsf{y}}|y')} \ell(\hat{y}, \hat{y}') < (2i-1)\varepsilon \right).
$$

The inequality (a) follows from the triangle inequality and (b) follows from the second part of the GS assumption noting that $\hat{y}'' \sim f_{\sharp} p_{\mathsf{x}|\mathsf{y}}(\cdot \mid y)$ and $\hat{y} \sim f_{\sharp} q_{\mathsf{x}|\mathsf{y}}^{(i)}(\cdot \mid y)$.

Finally, using that $i \leq d$ and $\varepsilon \leq \frac{M}{2d}$, we have that $(2i-1)\varepsilon < M$. This gives us

$$
\Pr_{\hat{y} \sim f_{\sharp} q_{\mathsf{x}|\mathsf{y}}^{(i)}(\cdot|y)} \left( \inf_{\hat{y}' \in \mathrm{supp}_f(p_{\mathsf{x}|\mathsf{y}}|y)} \ell(\hat{y}, \hat{y}') < (i-1)\varepsilon \right) \leq \Pr_{\hat{y} \sim f_{\sharp} p_{\mathsf{x}|\mathsf{y}}(\cdot|y)} \left( \inf_{\hat{y}' \in \mathrm{supp}_f(p_{\mathsf{x}|\mathsf{y}}|y)} \ell(\hat{y}, \hat{y}') < (2i-1)\varepsilon \right)
$$

$$
\leq \Pr_{\hat{y} \sim f_{\sharp} p_{\mathsf{x}|\mathsf{y}}(\cdot|y)} \left( \inf_{\hat{y}' \in \mathrm{supp}_f(p_{\mathsf{x}|\mathsf{y}}|y)} \ell(\hat{y}, \hat{y}') < M \right)
$$

$$
< \eta,
$$

where the last inequality follows from the PM assumption.

Together, this gives us

$$
\begin{aligned}
\Pr_{\hat{y}\sim f_\sharp q^{(i)}_{\mathrm{x|y}}(\cdot|y)} \left( \inf_{\hat{y}'\in\mathrm{supp}(f_\sharp p_\mathrm{x})} \ell(\hat{y},\hat{y}') < (i-1)\varepsilon \right) &\leq \sum_{y'\in\mathcal{Y}} \Pr_{\hat{y}\sim f_\sharp q^{(i)}_{\mathrm{x|y}}(\cdot|y)} \left( \inf_{\hat{y}'\in\mathrm{supp}_f(p_{\mathrm{x|y}}|y')} \ell(\hat{y},\hat{y}') < (i-1)\varepsilon \right) \\
&< \sum_{y':y\neq y'} \eta \\
&< |\mathcal{Y}|\eta.
\end{aligned}
$$

As required. $\qquad\square$

**Remark.** We note that Lemma A.1 can be improved in the $|\mathcal{Y}|$ dependency. Indeed, one can further refine this to be $|\{y' \mid y' \in \mathrm{supp}(p_\mathrm{y}), y' \neq y\}|$. This holds for all subsequent results, but for simplicity, we simply use $|\mathcal{Y}|$.

Lemma A.1 leads to an immediate result on the behavior of the worst case loss by using Markov's inequality.

**Corollary A.1.** *Under the same assumptions of Lemma A.1*

$$
\mathbb{E}_{\hat{y}\sim f_\sharp q^{(i)}_{\mathrm{x|y}}(\cdot|y)} \left[ \inf_{\hat{y}'\in\mathrm{supp}(f_\sharp p_\mathrm{x})} \ell(\hat{y},\hat{y}') \right] \geq (i-1)\varepsilon(1 - |\mathcal{Y}|\eta). \tag{15}
$$

Using the above result and applying the GS assumption, we can derive a lower bound on the marginal distance $W_{1,\ell}(p_\mathrm{x}, q_\mathrm{x})$.

**Lemma A.2.** *Given two joint distributions $p$ and $q$ and a classifier $f$ such that $p, f$ satisfy the $\mathrm{PM}(M,\eta)$ assumption and $p, q, f$ satisfy the $\mathrm{GS}(a, b, s, \varepsilon)$ assumption, then if $\varepsilon \leq \frac{M}{2d}$ for some $d > 1$*

$$
W_{1,\ell}(f_\sharp p_\mathrm{x}, f_\sharp q_\mathrm{x}) \geq \varepsilon(1 - |\mathcal{Y}|\eta) \sum_{i=1}^{d} r_i(i-1).
$$

*Proof.* We use the Kantorovich-Rubenstein duality

$$
W_{1,\ell}(f_\sharp p_\mathrm{x}, f_\sharp q_\mathrm{x}) = \sup_{g\in L_{1,\ell}} \mathbb{E}_{\hat{y}\sim f_\sharp q_\mathrm{x}} [g(\hat{y})] - \mathbb{E}_{\hat{y}\sim f_\sharp p_\mathrm{x}} [g(\hat{y})],
$$

where $L_{1,\ell}$ denotes the class of 1-Lipschitz functions w.r.t. the $\ell$ metric.

For convenience, we define a function $h(\hat{y}) \doteq \inf_{\hat{y}'\in\mathrm{supp}(f_\sharp p_\mathrm{x})} \ell(\hat{y},\hat{y}')$. It follows that

$$
h(\hat{y}) \leq \ell(\hat{y},\hat{y}') \leq \ell(\hat{y},\hat{y}'') + \ell(\hat{y}'',\hat{y}')
$$

for all $\hat{y}' \in \mathrm{supp}(f_\sharp p_\mathrm{x})$ and $\hat{y}'' \in \mathcal{Y}$. Taking $\hat{y}' \in \arg\min_{\hat{y}'\in\mathrm{supp}(f_\sharp p_\mathrm{x})} \ell(\hat{y}'',\hat{y}')$, we get

$$
h(\hat{y}) \leq \ell(\hat{y},\hat{y}'') + h(\hat{y}'') \implies h(\hat{y}) - h(\hat{y}'') \leq \ell(\hat{y},\hat{y}'')
$$

for all $\hat{y}, \hat{y}''$. That is, $h \in L_{1,\ell}$.

Thus, taking a lower bound on the Kantorovich-Rubenstein dual, we have

$$
\begin{aligned}
W_{1,\ell}(f_\sharp p_\mathrm{x}, f_\sharp q_\mathrm{x}) &\geq \mathbb{E}_{\hat{y} \sim f_\sharp q_\mathrm{x}}[h(\hat{y})] - \mathbb{E}_{\hat{y} \sim f_\sharp p_\mathrm{x}}[h(\hat{y})] \\
&\overset{(a)}{=} \mathbb{E}_{\hat{y} \sim f_\sharp q_\mathrm{x}}[h(\hat{y})] \\
&= \mathbb{E}_{y \sim q_\mathrm{y}} \mathbb{E}_{\hat{y} \sim (f_\sharp q_{\mathrm{x}|\mathrm{y}})(\cdot|y)}[h(\hat{y})] \\
&\overset{(b)}{=} \sum_{i=1}^{s} r_i \mathbb{E}_{y \sim q_\mathrm{y}} \mathbb{E}_{\hat{y} \sim (f_\sharp q_{\mathrm{x}|\mathrm{y}}^{(i)})(\cdot|y)}[h(\hat{y})] \\
&\geq \sum_{i=1}^{d} r_i \mathbb{E}_{y \sim q_\mathrm{y}} \mathbb{E}_{\hat{y} \sim (f_\sharp q_{\mathrm{x}|\mathrm{y}}^{(i)})(\cdot|y)}[h(\hat{y})] \\
&\overset{(c)}{\geq} \sum_{i=1}^{d} r_i \mathbb{E}_{y \sim q_\mathrm{y}}[(i-1)\varepsilon(1-|\mathcal{Y}|\eta)] \\
&= \varepsilon(1-|\mathcal{Y}|\eta) \sum_{i=1}^{d} r_i(i-1).
\end{aligned}
$$

The equality (a) follows from the identity of indiscernibles and noting that $h = 0$ inside $\mathrm{supp}(f_\sharp p_\mathrm{x})$. The inequality (b) follows from the first part of the GS assumption. Finally, the inequality (c) follows from Corollary A.1. $\qquad\square$

Next, we derive a bound similar to (2) of Lemma 2.1. Instead of having an additive label entanglement term, we pay a price of having a multiplicative constant on the source risk. We remind that the maximum label ratio is defined via $\rho \doteq \max_{y \in \mathcal{Y}} q_\mathrm{y}(y)/p_\mathrm{y}(y)$.

**Lemma A.3.** *Suppose the loss function $\ell \colon \mathcal{Y} \times \mathcal{Y} \to \mathbb{R}$ is a metric and that the classifier $f \colon \mathcal{X} \to \mathcal{Y}$ is surjective, then*

$$
\epsilon_q(f) \leq \rho\epsilon_p(f) + \mathbb{E}_{y \sim q_\mathrm{y}}\left[ W_{1,\ell}((f_\sharp p_{\mathrm{x}|\mathrm{y}})(\cdot \mid y), (f_\sharp q_{\mathrm{x}|\mathrm{y}})(\cdot \mid y)) \right].
$$

*Proof.* The proof follows from a slight modification of (Koç et al., 2025, Lemma 3.1)'s proof, but applied to conditional risks. Note the decomposition

$$
\epsilon_q(f) = \sum_{y \in \mathcal{Y}} q_\mathrm{y}(y) \mathbb{E}_{x \sim q_{\mathrm{x}|\mathrm{y}}(\cdot|y)}[\ell(f(x), y)].
$$

It follows (with slight modification of (Koç et al., 2025, Lemma 3.1)) that

$$
\mathbb{E}_{x \sim q_{\mathrm{x}|\mathrm{y}}(\cdot|y)}[\ell(f(x), y)] \leq \mathbb{E}_{x \sim p_{\mathrm{x}|\mathrm{y}}(\cdot|y)}[\ell(f(x), y)] + W_{1,\ell}((f_\sharp p_{\mathrm{x}|\mathrm{y}})(\cdot \mid y), (f_\sharp q_{\mathrm{x}|\mathrm{y}})(\cdot \mid y)).
$$

Thus, we have

$$
\begin{aligned}
\epsilon_q(f) &= \sum_{y \in \mathcal{Y}} q_\mathrm{y}(y) \mathbb{E}_{x \sim q_{\mathrm{x}|\mathrm{y}}(\cdot|y)}[\ell(f(x), y)] \\
&\leq \sum_{y \in \mathcal{Y}} q_\mathrm{y}(y) \left( \mathbb{E}_{x \sim p_{\mathrm{x}|\mathrm{y}}(\cdot|y)}[\ell(f(x), y)] + W_{1,\ell}((f_\sharp p_{\mathrm{x}|\mathrm{y}})(\cdot \mid y), (f_\sharp q_{\mathrm{x}|\mathrm{y}})(\cdot \mid y)) \right) \\
&\leq \left( \max_{y \in \mathcal{Y}} \frac{q_\mathrm{y}(y)}{p_\mathrm{y}(y)} \right) \sum_{y \in \mathcal{Y}} p_\mathrm{y}(y) \mathbb{E}_{x \sim p_{\mathrm{x}|\mathrm{y}}(\cdot|y)}[\ell(f(x), y)] + \sum_{y \in \mathcal{Y}} q_\mathrm{y}(y) W_{1,\ell}((f_\sharp p_{\mathrm{x}|\mathrm{y}})(\cdot \mid y), (f_\sharp q_{\mathrm{x}|\mathrm{y}})(\cdot \mid y)) \\
&= \rho\epsilon_p(f) + \sum_{y \in \mathcal{Y}} q_\mathrm{y}(y) W_{1,\ell}((f_\sharp p_{\mathrm{x}|\mathrm{y}})(\cdot \mid y), (f_\sharp q_{\mathrm{x}|\mathrm{y}})(\cdot \mid y)).
\end{aligned}
$$

As required. $\qquad\square$

Finally, we prove Theorem 3.1 by combining the above results: we will use the GS assumption together with the PM assumption to relate the conditional distances $W_{1,\ell}(p_{x|y}(\cdot \mid y), q_{x|y}(\cdot \mid y))$ to the marginal Wasserstein distance $W_{1,\ell}(p_x, q_x)$ for small enough gradual distance increments $\varepsilon$.

**Theorem A.1** (Theorem 3.1 restated). *Assume* $\mathrm{GS}(a, b, s, \varepsilon)$ *and* $\mathrm{PM}(M, \eta)$ *hold for densities* $p$ *and* $q$ *and a surjective classifier* $f$*; and loss* $\ell$ *is a metric. Then if* $1 - |\mathcal{Y}|\eta > 0$ *and there exists a* $d \in \mathbb{Z}_+$ *such that* $\varepsilon \leq \frac{M}{2d}$*, then*

$$\epsilon_q(f) \leq \rho\epsilon_p(f) + \frac{s(s+1)}{d(d-1)} \frac{b}{a(1 - |\mathcal{Y}|\eta)} W_{1,\ell}(f_\sharp p_x(x), f_\sharp q_x(x)).$$

*Proof.* We start by using Lemma A.3 and using the first part of the GS assumption.

$$\epsilon_q(f) \leq \rho\epsilon_p(f) + \mathop{\mathbb{E}}_{y \sim q_y} \left[ W_{1,\ell}((f_\sharp p_{x|y})(\cdot \mid y), (f_\sharp q_{x|y})(\cdot \mid y)) \right]$$

$$\stackrel{(a)}{\leq} \rho\epsilon_p(f) + \mathop{\mathbb{E}}_{y \sim q_y} \left[ \sum_{i=1}^{s} r_i W_{1,\ell}((f_\sharp p_{x|y})(\cdot \mid y), (f_\sharp q_{x|y}^{(i)})(\cdot \mid y)) \right]$$

$$\stackrel{(b)}{\leq} \rho\epsilon_p(f) + \mathop{\mathbb{E}}_{y \sim q_y} \left[ \sum_{i=1}^{s} r_i i \varepsilon \right]$$

$$= \rho\epsilon_p(f) + \sum_{i=1}^{s} r_i i \varepsilon$$

$$\stackrel{(c)}{\leq} \rho\epsilon_p(f) + \varepsilon b \sum_{i=1}^{s} i$$

$$= \rho\epsilon_p(f) + \varepsilon b \frac{s(s+1)}{2},$$

where $(a)$ holds from the first part of the GS assumption alongside the convexity of Wasserstein distance, $(b)$ follows from the first part of the GS assumption (we use the fact that both measures are conditioning on the same $y$), and $(c)$ follows from our upper bound on $r_i$ in GS.

We now bound $\varepsilon$ by Lemma A.2 and using the assumption that $1 - |\mathcal{Y}|\eta > 0$:

$$W_{1,\ell}(f_\sharp p_x, f_\sharp q_x) \geq \varepsilon(1 - |\mathcal{Y}|\eta) \sum_{i=1}^{d} r_i(i - 1)$$

$$\stackrel{(d)}{\implies} W_{1,\ell}(f_\sharp p_x, f_\sharp q_x) \geq \varepsilon(1 - |\mathcal{Y}|\eta) a \sum_{i=1}^{d} (i - 1)$$

$$\implies W_{1,\ell}(f_\sharp p_x, f_\sharp q_x) \geq \varepsilon(1 - |\mathcal{Y}|\eta) a \frac{d(d-1)}{2}$$

$$\implies \varepsilon \leq \frac{2}{d(d-1)} \frac{1}{(1 - |\mathcal{Y}|\eta) a} W_{1,\ell}(f_\sharp p_x, f_\sharp q_x), \tag{16}$$

where $(d)$ follows from our lower bound on $r_i$ in the GS assumption.

Hence, together we have

$$\epsilon_q(f) \leq \rho\epsilon_p(f) + \varepsilon b \frac{s(s+1)}{2}$$

$$\leq \rho\epsilon_p(f) + b \frac{s(s+1)}{2} \frac{2}{d(d-1)} \frac{1}{(1 - |\mathcal{Y}|\eta) a} W_{1,\ell}(f_\sharp p_x, f_\sharp q_x)$$

$$= \rho\epsilon_p(f) + \frac{s(s+1)}{d(d-1)} \frac{b}{a(1 - |\mathcal{Y}|\eta)} W_{1,\ell}(f_\sharp p_x, f_\sharp q_x).$$

As required. $\qquad\square$

We can actually improve the bound whenever the marginal Wasserstein distance drops below a certain small threshold.

**Corollary A.2.** *Suppose assumptions of Theorem A.1 hold. If $W_{1,\ell}(f_\sharp p_x, f_\sharp q_x) \leq \frac{M(1-|\mathcal{Y}|\eta)a(d-1)d}{4s}$,*

$$\epsilon_q(f) \leq \rho\epsilon_p(f) + \frac{b(s+1)}{a(1-|\mathcal{Y}|\eta)(s-1)}W_{1,\ell}(f_\sharp p_x, f_\sharp q_x).$$

*Proof.* Notice that if $s = d$, the result holds. As such, we will show that if Theorem A.1 holds for $d < s$ and our additional assumption, taking $s = d$ also satisfies the assumptions of Theorem A.1, i.e., $\varepsilon \leq M/(2s)$.

Suppose that $d < s$. From (16), we have that

$$\varepsilon \leq \frac{2W_{1,\ell}(f_\sharp p_x, f_\sharp q_x)}{(1-|\mathcal{Y}|\eta)ad(d-1)} \leq \frac{M}{2s}. \tag{17}$$

Hence, taking $d = s$ satisfies the needed assumption, as required. □

# B. WRR optimization for least squares

In this section we look into instantiations of the WRR bound (3) in the linear least squares case, assuming as usual the Gradual Shift (GS) and the Probabilistic Margin (PM) assumptions. We show throughout that this simple case is rich enough to elucidate the various ideas proposed in this work.

The optimization of the upper bound, assuming finite $\rho$ and $\lambda$[9], reduces to the WRR-optimization of $\mathcal{R}_\lambda(\beta, \Gamma)$. In the discrete case where $m$ labeled source inputs and $n$ unlabeled target inputs are available the WRR optimization can be written as

$$\min_{\beta, \Gamma} \quad \frac{1}{m}\sum_i (y_i - x_i^\mathrm{T}\beta)^2 + \frac{\lambda}{2}\sum_{i,j} \gamma_{i,j}(x_i^\mathrm{T}\beta - x_j'^{\mathrm{T}}\beta)^2, \tag{18}$$

$$\text{s.t.} \quad \Gamma 1 = \tfrac{1}{m}1, \tag{19}$$

$$\Gamma^\mathrm{T}1 = \tfrac{1}{n}1, \tag{20}$$

where we used the fact that the squared distance obeys the approximate triangle inequality with $\chi = 2$ (see Appendix D in Koç et al. (2025)). An alternating minimization (i.e., block-coordinate descent with blocks $\Gamma$ and $\beta$) approach to solve (18) can be seen as an *iterative least squares* approach where we iterate between solving the OT problem w.r.t. $\Gamma$, at each iteration $k$

$$\Gamma_{k+1} = \arg\min_{\Gamma \in \mathbb{R}_+^{m\times n}} \beta_k^\mathrm{T}\Big(\sum_{i,j}\gamma_{i,j}(x_i - x_j')(x_i - x_j')^\mathrm{T}\Big)\beta_k, \text{ s.t. } \Gamma 1 = \tfrac{1}{m}1, \Gamma^\mathrm{T}1 = \tfrac{1}{n}1, \tag{21}$$

and a regularized least squares regression given the coupling $\Gamma_k$, which can be solved by taking the derivative w.r.t. $\beta$ as in standard least squares

$$-\tfrac{1}{m}X_p^\mathrm{T}(y - X_p\beta_{k+1}) + \tfrac{\lambda}{2}\sum_{i,j}\gamma_{i,j}^{k+1}(x_i - x_j')(x_i - x_j')^\mathrm{T}\beta_{k+1} = 0,$$

where we ignored the (sub)differential of $\Gamma_{k+1}$ (with entries $\gamma_{i,j}^{k+1}$) w.r.t. the regression parameters $\beta$. The coupling-dependent term above can be rewritten as a *data-dependent regularizer*

$$\sum_{i,j}\gamma_{i,j}(x_i - x_j')(x_i - x_j')^\mathrm{T} = \begin{bmatrix}X_p\\X_q\end{bmatrix}^\mathrm{T}\underbrace{\begin{bmatrix}I/m & -\Gamma\\-\Gamma^\mathrm{T} & I/n\end{bmatrix}}_{M_{pq}(\Gamma)}\underbrace{\begin{bmatrix}X_p\\X_q\end{bmatrix}}_{X_{pq}}$$

---

[9]The parameter $\lambda$ depends on the unknown GS and PM parameters which are in general hard to estimate but an upper bound on $\lambda$ can be more easily estimated from data.

where $X_p, X_q$ store $x_i^{\mathrm{T}}$ and $x_j'^{\mathrm{T}}$ in each row, respectively. Denoting the set of constraints for optimal transport as $S \doteq \{\Gamma \in \mathbb{R}_{\geq 0}^{m \times n} | \Gamma 1 = \frac{1}{m}1, \Gamma^{\mathrm{T}}1 = \frac{1}{n}1\}$, we get the following iterations

$$\Gamma_{k+1} = \arg\min_{\Gamma \in S} \sum_{i,j} \gamma_{i,j}(x_i^{\mathrm{T}}\beta_k - x_j'^{\mathrm{T}}\beta_k)^2, \tag{22}$$

$$R_{k+1} = mX_{pq}^{\mathrm{T}}M_{pq}(\Gamma_{k+1})X_{pq}, \tag{23}$$

$$\beta_{k+1} = (X_p^{\mathrm{T}}X_p + \tfrac{\lambda}{2}R_{k+1})^{-1}X_p^{\mathrm{T}}y. \tag{24}$$

Note that the optimal transport in (22) is between one dimensional random variables $z \sim \beta_{k\sharp}p$ and $z' \sim \beta_{k\sharp}q$ and hence can be computed easily in closed form (by one-to-one matching of their quantiles). Equivalently, one can see the iteration (22) - (24) as an iterative search for the best (minimizing the WRR bound) direction for the sliced Wasserstein distance (Peyré & Cuturi, 2019).

**Local optima.** Unfortunately, the above WRR minimization framework can introduce *additional* local optima that was not there in the un-regularized source risk minimization (ERM) framework. Even for the least-squares regression case, we construct a simple example to show that the original convex problem becomes non-convex with two local optima, only one of which is a global optimum. Optimizers such as the alternating minimization approach described above will get stuck at the poor local optimum when initialized at a nearby point (that is in the basin of attraction).

Taking $m = n = 2$, $\lambda = 1$ for two-dimensional labeled inputs $x_1 = (a, 0), x_2 = (0, b)$ with labels $y_1 = -1$ and $y_2 = 1$ and unlabeled inputs $x_1' = (c, 0), x_2' = (0, d)$, we can see that the OT cost in (21) will be the minimum of two quadratic functions in $\beta = (\beta_1, \beta_2)^{\mathrm{T}}$

$$\tfrac{1}{2}\min\left\{(a-c)^2\beta_1^2 + (b-d)^2)\beta_2^2, (a^2+c^2)\beta_1^2 + (b^2+d^2)\beta_2^2\right\}, \tag{25}$$

which is *non-convex* w.r.t. $\beta$ and the source of the multiple local minima. The WRR objective in (18) will be given as

$$\mathcal{R}(\beta) = \tfrac{1}{2}\left((a\beta_1+1)^2 + (b\beta_2-1)^2 + \tfrac{1}{2}\min\left\{(a-c)^2\beta_1^2 + (b-d)^2\beta_2^2, (a^2+c^2)\beta_1^2 + (b^2+d^2)\beta_2^2\right\}\right),$$

which has two minima whenever the quadratics can each be activated in (25) for some $\beta$, i.e., the function $\chi(\beta) = ac\beta_1^2 + bd\beta_2^2$ crosses zero at some $(\beta_1, \beta_2)$. The minima can be obtained as (with some abuse of notation for the indices)

$$\beta_1^\star = \left(\frac{-2a}{3a^2 - 2ac + c^2}, \frac{2b}{3b^2 - 2bd + d^2}\right),$$

$$\beta_2^\star = \left(\frac{-2a}{3a^2 + c^2}, \frac{2b}{3b^2 + d^2}\right),$$

For instance for $a = 1, b = 4, c = -1, d = 2$, alternating descent at $\chi(\beta) > 0$ will converge to $\beta_1^\star = (-\frac{1}{3}, \frac{2}{9})$ with a WRR value of $\mathcal{R}(\beta_1^\star) = \frac{7}{18}$, while at $\chi(\beta) < 0$ will converge to $\beta_2^\star = (-\frac{1}{2}, \frac{2}{13})$ with a WRR value of $\mathcal{R}(\beta_2^\star) = \frac{23}{52}$.

Such constructions can be extended to cover standard neural networks trained with typical stochastic optimizers. To see this, consider a multi-layer perceptron (MLP) with ReLU activations: one can construct as above a *linear regime* with multiple local optima where all the activations are enabled. Note that entropy-regularization (typically using a small $\epsilon > 0$) will smoothen the basins of attraction of local optima in general but otherwise preserve their cardinality.

**Lower bound.** In addition to the local optima that can be observed during WRR optimization, we show here that the optimization performance can be arbitrarily *poor*, i.e., it is not guaranteed that the global optima have low value (thus weakening the practical implications of Theorem 3.1). We use the fact that the 2-Wasserstein distance between two distributions $p$ and $q$ is lower bounded by the norm of the difference between their means $\Delta\mu = \mu - \mu'$, i.e., in our case

$$W_2^2(\beta_\sharp p_x, \beta_\sharp q_x) \geq \left(\beta^{\mathrm{T}}\Delta\mu\right)^2. \tag{26}$$

This means the WRR objective (18) is lower bounded by

$$
\begin{aligned}
\mathcal{R}(\beta, \Gamma) &\geq \tfrac{1}{m}\|y - X_p\beta\|^2 + \tfrac{\lambda}{2}\beta^{\mathrm{T}}\Delta\mu\Delta\mu^{\mathrm{T}}\beta, \\
&= \tfrac{1}{m}y^{\mathrm{T}}y - \tfrac{2}{m}\beta^{\mathrm{T}}X_p^{\mathrm{T}}y + \beta^{\mathrm{T}}H\beta, && (\text{defined } H \doteq \tfrac{1}{m}X_p^{\mathrm{T}}X_p + \tfrac{\lambda}{2}\Delta\mu\Delta\mu^{\mathrm{T}}) \\
&\geq \tfrac{1}{m}y^{\mathrm{T}}y - \tfrac{1}{m}\beta^{\star\mathrm{T}}X_p^{\mathrm{T}}y, && (\text{plugging in minimizer } H\beta^{\star} = \tfrac{1}{m}X_p^{\mathrm{T}}y), \\
&= \tfrac{1}{m}y^{\mathrm{T}}y - \tfrac{1}{m}b^{\mathrm{T}}H^{-1}b, && (\text{defined } b \doteq X_p^{\mathrm{T}}y), \\
&= \tfrac{1}{m}y^{\mathrm{T}}y - \tfrac{1}{m}b^{\mathrm{T}}(X_p^{\mathrm{T}}X_p)^{-1}b - \tfrac{1}{m}b^{\mathrm{T}}(H^{-1} - (X_p^{\mathrm{T}}X_p)^{-1})b, && (\text{comparing to ERM solution}), \\
&= \tfrac{1}{m}\|y - X_p\beta_{\mathrm{erm}}\|^2 - \tfrac{1}{m}b^{\mathrm{T}}(H^{-1} - (X_p^{\mathrm{T}}X_p)^{-1})b, && (\text{plugging in } \beta_{\mathrm{erm}} = (X_p^{\mathrm{T}}X_p)^{-1}b), \\
&= \tfrac{1}{m}\|y - X_p\beta_{\mathrm{erm}}\|^2 + \tfrac{\lambda}{2m}\frac{b^{\mathrm{T}}(X_p^{\mathrm{T}}X_p)^{-1}\Delta\mu\Delta\mu^{\mathrm{T}}(X_p^{\mathrm{T}}X_p)^{-1}b}{1 + \tfrac{\lambda}{2}\Delta\mu^{\mathrm{T}}(X_p^{\mathrm{T}}X_p)^{-1}\Delta\mu}, && (\text{rank-1 inverse formula}), \\
&= \tfrac{1}{m}\|y - X_p\beta_{\mathrm{erm}}\|^2 + \tfrac{\lambda}{2m}\frac{(\beta_{\mathrm{erm}}^{\mathrm{T}}\Delta\mu)^2}{1 + \tfrac{\lambda}{2}\Delta\mu^{\mathrm{T}}(X_p^{\mathrm{T}}X_p)^{-1}\Delta\mu}, && (\text{plugging in } \beta_{\mathrm{erm}}),
\end{aligned}
$$

where in the second to last step we applied the Sherman-Morrison formula for the rank-1 inverse $(A + \lambda vv^{\mathrm{T}})^{-1} = A^{-1} - \frac{\lambda A^{-1}vv^{\mathrm{T}}A^{-1}}{1 + \lambda v^{\mathrm{T}}A^{-1}v}$.

The second term of the lower bound that includes $\beta_{\mathrm{erm}}^{\mathrm{T}}\Delta\mu$ indicates that mean shifts that are aligned with the optimal ERM parameter can raise the values of the WRR optima. For example, in the previous example with two points, plugging in $a = 1, b = 4, c = -1, d = 2$, the source risk evaluates to zero at $\beta_{\mathrm{erm}} = (-1/a, 1/b)^{\mathrm{T}}$ and the second term evaluates for $\lambda = 1$ to $9/98 \leq \min(\mathcal{R}(\beta_1^{\star}), \mathcal{R}(\beta_2^{\star}))$.

Note that one can sharpen the above by using the Gelbrich lower bound (Gelbrich, 1990), which in this case, given source and target input covariance matrices $\Sigma, \Sigma'$ becomes $(\beta^{\mathrm{T}}\Delta\mu)^2 + (\sqrt{\beta^{\mathrm{T}}\Sigma\beta} - \sqrt{\beta^{\mathrm{T}}\Sigma'\beta})^2$. Finally, whenever label shift is present, the lower bound in Lemma 3.1 suggests that any lower bound that we derive on the WRR objective would be increased by the amount of label shift $W_{1,\ell}(p_y, q_y)$.

**Gradual Shift.** We explore the implications of Theorem 3.1 here in an example. In particular, we want to quantify the conclusion that *low WRR implies low target risk* whenever the GS and PM assumptions hold uniformly for $f_{\sharp}p$ and $f_{\sharp}q$, e.g., for all models that separate the source conditionals sufficiently well. Although we showed above that in general, WRR optimization is non-convex and that there are potentially poor local minima (depending on the shift and the model) which could hinder successful minimization of the objective, assuming the optimizer can find the global optimum with a low WRR value, the target risk will be also low (depending on the shift multiplier $\lambda$).

We take as before the linear least squares problem and analyze the *population* target risk with labels $y_1 = -1$ and $y_2 = 1$. We assume for simplicity that Gaussian class-conditionals $p(x \mid y_i) = \mathcal{N}(\mu_i, \Sigma_i)$, $i = 1, 2$ *shift in input space* $x \in \mathcal{R}^d, d > 1$ to $q(x \mid y_i) = \mathcal{N}(\mu_i', \Sigma_i')$ with equal mass (hence $W_1(p_y, q_y) = 0$). The covariance of the Gaussians and the intersection of their supports correlate with the intensity of gradual shift, e.g., a wide $\Sigma_i'$ with a reasonably small mean shift $\mu_i' - \mu_i$ roughly corresponds to observing various shifts $q^{(i)}$ with small shift intensity $\varepsilon$ between them. In this case, the target bound (setting $\chi = 2$ as before) becomes

$$
\epsilon_q(\beta) \leq 4\epsilon_p(\beta) + 2\lambda(\beta)W_2^2(\beta_{\sharp}p_x, \beta_{\sharp}q_x),
$$

where $\lambda(\beta)$ depends on the observable intensity of the shift in output space. It is difficult to analyze Wasserstein distances between *mixtures* of Gaussians $\beta_{\sharp}p = \sum_{i=1,2}\alpha_i\mathcal{N}(\beta^{\mathrm{T}}\mu_i, \beta^{\mathrm{T}}\Sigma_i\beta)$ and $\beta_{\sharp}q = \sum_{i=1,2}\alpha_i\mathcal{N}(\beta^{\mathrm{T}}\mu_i', \beta^{\mathrm{T}}\Sigma_i'\beta)$ such that $\sum_{i=1,2}\alpha_i = 1$ directly, hence we use instead the upper bound using $MW_2$, a Wasserstein-type distance between two Gaussian mixtures (Chen et al., 2018; Delon & Desolneux, 2020)

$$
W_2^2(p, q) \leq MW_2^2(p, q),
$$

whose computation for 2-mixtures reduces to the 2-by-2 case, but with modified costs given by the $W_2^2$ distance between

Gaussians, which can be computed in closed form

$$W_2^2\big(\mathcal{N}(\mu_1, \Sigma_1), \mathcal{N}(\mu_2, \Sigma_2)\big) = \|\mu_1 - \mu_2\|^2 + \mathcal{B}(\Sigma_1, \Sigma_2)^2,$$
$$\mathcal{B}(\Sigma_1, \Sigma_2)^2 = \mathrm{tr}\Big(\Sigma_1 + \Sigma_2 - 2(\Sigma_1^{1/2}\Sigma_2\Sigma_1^{1/2})^{1/2}\Big),$$
$$= 2\mathrm{tr}\Big(\mathrm{AM}(\Sigma_1, \Sigma_2) - \mathrm{GM}(\Sigma_1, \Sigma_2)\Big).$$

The term $\mathcal{B}(\Sigma_1, \Sigma_2)$ is the *Bures* metric between two positive semidefinite (psd) matrices and is equal to two times the trace of the *arithmetic-geometric mean gap* $\mathrm{AM}(\Sigma_1, \Sigma_2) - \mathrm{GM}(\Sigma_1, \Sigma_2)$. The closed form above can be simplified for one-dimensional Gaussians to

$$W_2^2\big(\mathcal{N}(\mu_1, \sigma_1^2), \mathcal{N}(\mu_2, \sigma_2^2)\big) = (\mu_1 - \mu_2)^2 + (\sigma_1 - \sigma_2)^2.$$

Applying it to $MW_2^2(\beta_\sharp p_x, \beta_\sharp q_x)$ and defining the following two matrices, corresponding to the cases where we *match* or *swap* the shifting class-conditional distributions during OT assignments

$$M_{\mathrm{match}} \doteq \sum_{i=1,2} (\mu_i - \mu_i')(\mu_i - \mu_i')^{\mathrm{T}} + 2\Big(\mathrm{AM}(\Sigma_i, \Sigma_i') - \mathrm{GM}(\Sigma_i, \Sigma_i')\Big),$$

$$M_{\mathrm{swap}} \doteq \sum_{i,j=1,2|i\neq j} (\mu_i - \mu_j')(\mu_i - \mu_j')^{\mathrm{T}} + 2\Big(\mathrm{AM}(\Sigma_i, \Sigma_j') - \mathrm{GM}(\Sigma_i, \Sigma_j')\Big),$$

we get the following upper bound

$$MW_2^2(\beta_\sharp p_x, \beta_\sharp q_x) = \min\Big\{ \beta^{\mathrm{T}}\Big(\textstyle\sum_{i=1,2}(\mu_i - \mu_i')(\mu_i - \mu_i')^{\mathrm{T}}\Big)\beta + \big(\sqrt{\beta^{\mathrm{T}}\Sigma_1\beta} - \sqrt{\beta^{\mathrm{T}}\Sigma_1'\beta}\big)^2 + \big(\sqrt{\beta^{\mathrm{T}}\Sigma_2\beta} - \sqrt{\beta^{\mathrm{T}}\Sigma_2'\beta}\big)^2,$$
$$\beta^{\mathrm{T}}\Big(\textstyle\sum_{i\neq j}(\mu_i - \mu_j')(\mu_i - \mu_j')^{\mathrm{T}}\Big)\beta + \big(\sqrt{\beta^{\mathrm{T}}\Sigma_1\beta} - \sqrt{\beta^{\mathrm{T}}\Sigma_2'\beta}\big)^2 + \big(\sqrt{\beta^{\mathrm{T}}\Sigma_1'\beta} - \sqrt{\beta^{\mathrm{T}}\Sigma_2\beta}\big)^2\Big\},$$
$$\leq \min\Big\{ \beta^{\mathrm{T}} M_{\mathrm{match}}\beta, \beta^{\mathrm{T}} M_{\mathrm{swap}}\beta \Big\},$$

where we used the (tight) lower bound on the square-root terms $\sqrt{\beta^{\mathrm{T}}\Sigma_1\beta\beta^{\mathrm{T}}\Sigma_2\beta} \geq \beta^{\mathrm{T}}\mathrm{GM}(\Sigma_1, \Sigma_2)\beta$ for any two psd matrices $\Sigma_1, \Sigma_2$.

We want to show that under the assumptions of Theorem 3.1, we cannot have a situation where the right hand side of

$$\epsilon_q(\beta) \leq \mathcal{R}_\lambda(\beta) \leq \bar{\mathcal{R}}_\lambda(\beta) \leq \min\Big\{ \tfrac{1}{2}\beta^T\Big(\mu_1\mu_1^{\mathrm{T}} + \mu_2\mu_2^{\mathrm{T}} + \Sigma_1 + \Sigma_2 + \lambda M_{\mathrm{match}}\Big)\beta + (\mu_1 - \mu_2)^{\mathrm{T}}\beta + 1,$$
$$\tfrac{1}{2}\beta^T\Big(\mu_1\mu_1^{\mathrm{T}} + \mu_2\mu_2^{\mathrm{T}} + \Sigma_1 + \Sigma_2 + \lambda M_{\mathrm{swap}}\Big)\beta + (\mu_1 - \mu_2)^{\mathrm{T}}\beta + 1\Big\},$$

is low, but the target risk is high. Consider the anti-symmetric case where the source distribution with parameters $\mu_1 = \mu, \mu_2 = -\mu$, and $\Sigma_1 = \Sigma_2 = \Sigma$ shifts to the target distribution with parameters $\mu_1' = \mu + \nu, \mu_2' = -\mu - \nu$ and $\Sigma_1' = \Sigma_2' = S$ for some psd matrix $S$ big enough to satisfy gradual shift. In this case the WRR upper bound evaluates to

$$\bar{\mathcal{R}}_\lambda(\beta) = 4(\beta^{\mathrm{T}}\mu + 1)^2 + 4\beta^{\mathrm{T}}\Sigma\beta + \lambda\min\{\beta^{\mathrm{T}} M_{\mathrm{match}}\beta, \beta^{\mathrm{T}} M_{\mathrm{swap}}\beta\},$$
$$= 4(\beta^{\mathrm{T}}\mu + 1)^2 + 4\beta^{\mathrm{T}}\Sigma\beta + \lambda\min\{(\beta^{\mathrm{T}}\nu)^2 + \beta^{\mathrm{T}}\Delta\beta, (\beta^{\mathrm{T}}(2\mu + \nu))^2 + \beta^{\mathrm{T}}\Delta\beta\},$$

where we defined $\Delta \doteq \mathrm{AM}(\Sigma, S) - \mathrm{GM}(\Sigma, S)$. Whenever $\beta^{\mathrm{T}}\nu$ is small enough and the PM assumptions holds for some $M > 0$ (not too low), $(\beta^{\mathrm{T}}\nu)^2 < (\beta^{\mathrm{T}}(2\mu + \nu))^2$ will be satisfied. Then given $\beta^{\mathrm{T}} S\beta < \alpha\beta^{\mathrm{T}}\Sigma\beta$ for some small $\alpha$, indicating the requirement of GS for this case, a small $\bar{\mathcal{R}}_\lambda(\beta)$ will force the target risk

$$\epsilon_q(\beta) = \big((\beta^{\mathrm{T}}\mu + 1) + \beta^{\mathrm{T}}\nu\big)^2 + \beta^{\mathrm{T}} S\beta,$$
$$= (\beta^{\mathrm{T}}\mu + 1)^2 + (\beta^{\mathrm{T}}\nu)^2 + 2(\beta^{\mathrm{T}}\mu + 1)(\beta^{\mathrm{T}}\nu) + \beta^{\mathrm{T}} S\beta,$$

to be small as well, since $(\beta^{\mathrm{T}}\mu + 1)$ and $\beta^{\mathrm{T}} S\beta$ terms are both small.

## B.1. Weighted WRR optimization.

In the case where we tighten the WRR-bound in (18) by looking for the best source weighting, the entropy-regularized problem becomes

$$\min_{\beta, w, \Gamma} \quad \sum_i w_i (y_i - x_i^{\mathrm{T}} \beta)^2 + \frac{\lambda}{2} \sum_{i,j} \gamma_{i,j} \big( (x_i^{\mathrm{T}} \beta - x'^{\mathrm{T}}_j \beta)^2 + \epsilon \log \gamma_{i,j} \big),$$

$$\text{s.t.} \quad \Gamma 1 = w,$$

$$\Gamma^{\mathrm{T}} 1 = \tfrac{1}{n} 1.$$

As described in Section 4, the optimal semi-relaxed coupling for the problem above is a column-wise softmax-distribution. Plugging the solution to the above problem, we get the following unconstrained optimization problem

$$\min_{\beta} \frac{1}{n} \sum_{i,j} \frac{\exp(-\tilde{C}_{ij}/\epsilon)}{\sum_k \exp(-\tilde{C}_{kj}/\epsilon)} \tilde{C}_{ij},$$

where $\tilde{C}_{ij}(\beta) = C_{ij} + \frac{2}{\lambda}(y_i - x_i^{\mathrm{T}} \beta)^2$ and the pairwise costs are $C_{ij} = (x_i^{\mathrm{T}} \beta - x'^{\mathrm{T}}_j \beta)^2$.

**Alternating minimization.** As before, if we omit the derivatives of the coupling w.r.t. $\beta$, we can use alternating minimization to solve

$$-X_p^{\mathrm{T}} W (y - X_p \beta_k) + \frac{\lambda}{2} \sum_{i,j} \gamma_{i,j} (x_i - x'_j)(x_i - x'_j)^{\mathrm{T}} \beta_k = 0,$$

where the derived weight matrix is $W = \mathrm{diag}\big(\sum_j \gamma_{ij}\big)$. The above equation can be organized as before to yield a *weighted* version of the data-dependent regularizer

$$\sum_{i,j} \gamma_{i,j} (x_i - x'_j)(x_i - x'_j)^{\mathrm{T}} = \begin{bmatrix} X_p \\ X_q \end{bmatrix}^{\mathrm{T}} \underbrace{\begin{bmatrix} W & -\Gamma \\ -\Gamma^{\mathrm{T}} & I/n \end{bmatrix}}_{\bar{M}_{pq}(\Gamma)} \begin{bmatrix} X_p \\ X_q \end{bmatrix},$$

and the resulting *iteratively weighted least squares* iterations are given as

$$\Gamma_{k+1} = \frac{1}{n} \frac{\exp(-\tilde{C}_{ij}(\beta_k)/\epsilon)}{\sum_l \exp(-\tilde{C}_{lj}(\beta_k)/\epsilon)}, \tag{27}$$

$$W_{k+1} = \mathrm{diag}\big(\sum_j \gamma_{ij}^{k+1}\big), \tag{28}$$

$$\bar{R}_{k+1} = X_{pq}^{\mathrm{T}} \bar{M}_{pq}(\Gamma_{k+1}) X_{pq}, \tag{29}$$

$$\beta_{k+1} = (X_p^{\mathrm{T}} W_{k+1} X_p + \lambda \bar{R}_{k+1})^{-1} X_p^{\mathrm{T}} W_{k+1} y. \tag{30}$$

Note that the previous iterations (22) - (24) can be written in the same form with *flat weighting*, i.e., $W = I/m$.

**Nonlinear equations.** Including the derivative of the entropy-regularized and hence differentiable coupling w.r.t $\beta$, we get a variant of the above iterations, which can be interpreted as solving the resulting nonlinear equations w.r.t. $\beta$ iteratively

$$-X_p^{\mathrm{T}} \tilde{W} (y - X_p \beta_k) + \frac{\lambda}{2} \sum_{i,j} \tilde{\gamma}_{i,j} (x_i - x'_j)(x_i - x'_j)^{\mathrm{T}} \beta_k = 0,$$

where $\tilde{\gamma}_{ij}$ is a multiplicative perturbation of $\gamma_{ij}$

$$\tilde{\gamma}_{ij} = \gamma_{ij}(1 + \delta_{ij}), \ \delta_{ij} = \frac{n \sum_k \gamma_{kj} \tilde{C}_{kj} - \tilde{C}_{ij}}{\epsilon},$$

and the derived weights are modified to $\tilde{W} = \mathrm{diag}(\sum_i \tilde{\gamma}_{ij})$. The entries $\tilde{\gamma}_{ij}$ can be organized into a matrix form $\tilde{\Gamma}$

$$\tilde{\Gamma} = \Gamma \cdot (1 + \Delta),$$
$$\Delta = \frac{\bar{c}1^{\mathrm{T}} - \tilde{C}}{\epsilon},$$
$$\bar{c}_j = n \sum_i \gamma_{ij} \tilde{C}_{ij}.$$

Note that $\bar{c}_j$ represents the average optimal transport component cost of moving the point $x'_j$. The resulting iterations can then be obtained by substituting $\tilde{\Gamma}$ instead of $\Gamma$ above

$$\Gamma_{k+1} = \frac{1}{n} \frac{\exp(-\tilde{C}_{ij}(\beta_k)/\epsilon)}{\sum_l \exp(-\tilde{C}_{lj}(\beta_k)/\epsilon)},$$
$$\tilde{\Gamma}_{k+1} = \Gamma_{k+1} + \frac{n}{\epsilon}\Gamma_{k+1}\mathrm{diag}(\sum_i \gamma_{ij}^{k+1}\tilde{C}_{ij}(\beta_k)) - \frac{1}{\epsilon}\Gamma_{k+1} \cdot \tilde{C}(\beta_k),$$
$$R_{k+1} = X_{pq}^{\mathrm{T}} M_{pq}(\tilde{\Gamma}_{k+1}) X_{pq},$$
$$\tilde{W}_{k+1} = \mathrm{diag}(\sum_i \tilde{\gamma}_{ij}^{k+1}),$$
$$\beta_{k+1} = (X_p^{\mathrm{T}} \tilde{W}_{k+1} X_p + \lambda R_{k+1})^{-1} X_p^{\mathrm{T}} \tilde{W}_{k+1} y.$$

**Lower bound.** Unlike the vanilla WRR optimization, in the weighted WRR optimization, whenever there is no label shift, the lower bound in (26) becomes trivially zero: the optimizer can weight the source (at the expense of some additional variance) to deflate the mean shift. When there is label shift, the problem becomes more complex: the bound in (5) suggests that a weighting that fortuitously keeps $W_{1,\ell}(p_y^w, q_y)$ minimal would be able to lower the target risk, while another weighting could also inflate the label shift and hence suffer from high target risk (and incur a high value of the $\mathrm{W}^2\mathrm{R}^2$ objective as well).

## C. More Related Work

We continue the discussion from the main text on related work, describing in more detail the connections to the domain adaptation (DA) literature.

**Importance Weighting.** Importance weighting (IW) has been considered traditionally in the input space (Shimodaira, 2000), unlike our setting which is in the feature space (typically the outputs of a neural network). Under the *covariate shift* assumption (where $p_{y|x}$ is assumed to be invariant across environments) it can be shown that the standard IW estimation is consistent. Efficient IW estimators have been constructed using, e.g., convex programs (Huang et al., 2006; Sugiyama et al., 2007) that directly estimate the density ratio of the source and target distributions. These objectives by Huang et al. (2006); Sugiyama et al. (2007) can be shown to minimize the MMD-distance and KL-divergence between the source and target distributions, respectively, under the *shared support* assumption (where $\mathrm{supp}(q) \subseteq \mathrm{supp}(p)$).

For the *out-of-support* case, weighted IPM bounds for DA appeared in Johansson et al. (2022) in a general causal inference context, where the authors restricted the learned representations to invertible functions in order to limit the effect of the incomputable conditional discrepancies appearing in the bounds. The authors used MMD as the discrepancy of choice and estimated the importance weights using a separate neural network. A typical sample complexity bound were included, which highlighted the importance of regularizing the source weighting, as a result of the inherent bias-variance trade-off. As a clear distinction to our setting, note that the analysis presented in Johansson et al. (2022) does not lead to an algorithmic approach, unlike the case in Theorem 3.1 (and extensions), which leads to the unbalanced OT as a relaxed alignment strategy.

**Unbalanced Optimal Transport.** Our algorithm that tightens the Wasserstein based OT bound (Koç et al., 2025) via importance weighting is also related to distributionally robust optimization (DRO) which uses Sinkhorn distances (Wang et al., 2025): Sinkhorn DRO in particular reduces to the semi-relaxed OT that we use for domain adaptation whenever the reference distribution $\nu$ used in the entropy regularization (Wang et al., 2025) is set as the target distribution $q$. In the absence of such unlabeled domain information, the reference distribution will be typically set to the empirical source distribution $\hat{p}$, which will revert the algorithm to weighted ERM.

Considering a weighted distribution as an argument of a Wasserstein domain matching regularizer has previously been explored (Wu et al., 2019). In this prior work, the proposed relaxation is based on achieving $\beta$-admissibility. The main difference to our work is that their approach (without explicitly mentioning) solves a variant of *partial* optimal transport (Figalli, 2010) (POT), where instead of dropping the first marginal constraint in (6), the equality constraint is replaced to an inequality, i.e., $\Gamma^{\mathrm{T}} \mathbb{1}_n \leq (1 + \alpha) \mathbb{1}_m$ for some $\alpha > 0$.

Unbalanced Optimal Transport (UOT), as mentioned in the main text, was considered as a more robust alignment strategy that can diminish the effects of outliers during domain adaptation and was shown to fare better when estimated via minibatching (Fatras et al., 2021). This idea was extended in Nguyen et al. (2022b) by switching to a POT formulation, which was argued to make hyperparameter selection of the penalty scales more robust to cost scales. In contrast to these two approaches, we apply weighting directly on the WRR objective and derive a variant UOT problem that includes weighted source loss as a component. This implies that, unlike the standard UOT setting where the cost matrices are calculated between feature (source and target) points, points with high source loss would be less likely to be upweighted during the UOT optimization.

**Assumptions.** The same authors in Wu et al. (2019) make a *connectedness* assumption to justify their relaxed alignment procedure. Although conceptually related to our GS assumption, the definition of the connectedness is quite different in various ways: (a) it is assumed that the source and target conditionals are connected in the input space and (b) unlike our setting, the target conditionals need to be contiguous, i.e., next to each other (so that one can trace a path from the source to the target distributions with high probability). Making the connectedness assumption in the input space is prohibitive for several reasons: (1) it is difficult to satisfy in most datasets with high dimensional input spaces and equally difficult to observe from (finite) samples that typically do not cover the space adequately, (2) it requires one to make an additional Lipschitzness assumption, which is difficult to satisfy for deep neural networks without making the Lipschitz constant prohibitively large, thereby potentially shrinking the connectedness in the latent space massively and hampering its observability.

As for our Probabilistic Margin (PM) assumption, it was inspired by the Probabilistic Lipschitzness assumption explored in the literature (Ben-David & Urner, 2014; Courty et al., 2017).

## D. Experimental Details

In this section we give more details about our experimental setup that was discussed briefly in section 5 of the main text.

**Datasets.** MNIST and USPS are $28 \times 28$ and $16 \times 16$ grayscale datasets with similar looking digits, while SVHN is a $32 \times 32$ colored street house number dataset where the label corresponds to the middle digit. MNIST-M is an *adversarial* dataset where the MNIST digits are laid over a colored background corresponding to another dataset (Ganin et al., 2016). In our experiments we downsample the MNIST digits to $16 \times 16$ in the USPS scenarios, and in the SVHN to MNIST and MNIST to MNIST-M scenarios, we duplicate the MNIST dataset, increasing the number of channels to three.

**Methods.** We compare our proposed approach to various promising state-of-the-art DA methods used in the literature. While we only included three other methods, DANN (Ganin et al., 2016), reverse-KL (Nguyen et al., 2022a) and WRR (Courty et al., 2017; Koç et al., 2025) in Table 1 due to space constraints, we compared our approach against three others as well: (deep) JDOT (Courty et al., 2017; Damodaran et al., 2018), MMD (Long et al., 2015; Gulrajani & Lopez-Paz, 2021) and f-DAL (Acuna et al., 2021). The test accuracies of these methods are shown in Table 2. Below we discuss these methods and mention various issues in optimizing their objectives.

The *domain adversarial neural networks* (DANN) approach (Ganin et al., 2016) uses an adversarial approach, inspired by the $H \Delta H$ divergence proposed in Ben-David et al. (2010), to optimize, in addition to the source classifier, a discriminator network that learns to separate the source from the target feature distributions. Rather than approximating the $H \Delta H$ divergence, it was shown by Acuna et al. (2021) that the method approximates Jensen-Shannon (JS) divergence instead. We find in our experiments that the model architecture (e.g., MLP or a ConvNet) chosen for the discriminator, as well as the learning rates, is essential to a stable implementation of DANN.

The adversarial framework originally proposed in DANN was later extended and generalized to $f$-divergences, and the resulting method, *f-DAL* (Acuna et al., 2021), in distinction to DANN, proposes to use a discriminator that has the same

architecture as the classifier.[10] We find in our experiments that this choice of model architecture (i.e., duplicating the classifier with re-initialized weights) often ends in *unstable* learning configurations. The source of these instabilities is the $f$-divergence itself: in the *out-of-source-support* cases that we often find in standard DA benchmarks, the variational lower bound of the $f$-divergence optimized easily becomes unbounded whenever the discriminator is not seriously constrained (e.g., using a different model with bounded outputs, severely capping the gradient norms, etc.) It appears that such adversarial methods are promising only to the extent that they *do not* approximate the $f$-divergences accurately. We use a small convolutional discriminator (same as DANN) together with a small learning rate of $1 \times 10^{-4}$ to stabilize f-DAL, but we still find significant instabilities that increase variance and make the learning curves non-monotonic.

The *reverse-KL* method proposed in Nguyen et al. (2022a) applies, unlike what its name suggests, the forward as well as reverse KL divergence as the discrepancy term using Probabilistic Representation Networks (Kendall & Gal, 2017). KL-divergences can be easily computed in this probabilistic framework by *sampling* from the network, whose optimized weights are interpreted as the parameters of a multivariate Gaussian distribution (with diagonal covariance matrices). Whenever the hyperparameters of this approach are tuned in between the tested scenarios, this algorithm performs very well (see Table 2 in Nguyen et al. (2022a)) and is clearly a promising approach. We thus implement probabilistic versions of the models mentioned in Section 5, by doubling the size of the penultimate layer, and then sampling from the distribution parameterized by that layer's outputs (mean and standard deviations being set to the first and second half of the layer outputs, respectively.) As can be seen from the results in Table 1, we find that such networks are difficult to train and that their performance is excessively sensitive to the specification of hyperparameters (see, e.g., ablations in (Koç et al., 2025)).

WRR can be seen as a *barebones* version of the joint distribution optimal transportation (JDOT) approach proposed in Courty et al. (2017) and later extended to deep networks in Damodaran et al. (2018). In distinction to WRR, whose domain alignment we implement only in the *output layer*, implementations of JDOT typically add a (scaled) feature layer alignment as well. The idea of aligning the distribution at multiple layers is promising, however it is not clear how to scale the different layer alignments appropriately. In our experiments, we find that there does not seem to be an easy rule to guarantee high accuracy across multiple scenarios: JDOT can in a few cases improve over WRR accuracy, but lags behind WRR in others.

Finally, for the MMD method, we used an MMD-based implementation from the *DomainBed* repository (Gulrajani & Lopez-Paz, 2021) which uses a mixture of different kernels during the domain alignment. The mixture-based approach requires a more elaborate setting of the hyperparameters, and while we think that MMD is a promising approach (with a smoother derivative), we have not found a setting that works better than $W^2R^2$ in any of the DA experiments.

We think that the optimization issues of the methods (i.e., training their models) highlighted above can be alleviated by a deeper investigation and tightening of the DA bounds, which this research represents. In particular, a more precise characterization of the scales (e.g., $\lambda(w)$ in Corollary 4.1) appearing in the bounds, including their dependence on the model and distribution shift characteristics (as, e.g., analyzed using our GS and PM assumptions), will enable new, more flexible DA approaches that can auto-tune the divergences used during the optimization.

**Oracles.** Besides the methods mentioned above, we continue the recent trend to include oracles (Koç et al., 2025) and include two different oracles in each experiment that are allowed access to target labels. The *low-joint-error* (LJE) oracle minimizes the sum of source and target empirical risk, while the *close-conditionals* (CC) oracle minimizes the *joint* Wasserstein regularized risk $\epsilon_p(f) + W_{1,\ell}(f_\sharp p, f_\sharp q)$, where $p(x, y)$ and $q(x, y)$ are the joint source and target distributions, respectively.

Both oracles act as the *ceiling* or asymptotic limit of what we can expect in a successful DA experiment. Some models cannot be optimized effectively in various scenarios (e.g., MLP performs poorly in SVHN scenarios) and oracles are convenient tools that we use to indicate and debug such problems. Besides the LJE oracle, the CC-oracle acts as a tighter indicator of model capacity for domain adaptation: (depending on the model and scenario considered) the LJE oracle often *spreads* the source and target output conditionals unlike the CC, which is tasked with the more stringent task of aligning the joint distributions.

---

[10] Although all $f$-divergences are supported, the authors in Acuna et al. (2021) report $\chi^2$-divergence as the best performing discrepancy term.

*Table 2.* Test accuracy results of UDA algorithms, including *f-DAL*, *MMD* and *JDOT*, complementing the results shown in Table 1. The results, as before, are evaluated over three different models and various UDA scenarios. For the performance of oracles and ERM, not shown here, please see Table 1. Hyperparameters are fixed across different scenarios to create a more difficult but realistic evaluation setting. Accuracy and s.t.d. is reported over five trials. The best performing algorithm is boxed.

| Scenario | Model | rev-KL | DANN | f-DAL | MMD | JDOT | WRR | $W^2R^2$ |
|---|---|---|---|---|---|---|---|---|
| MNIST → USPS | MLP | $0.62 \pm 0.011$ | $0.55 \pm 0.054$ | $0.67 \pm 0.0407$ | $0.77 \pm 0.0216$ | $0.75 \pm 0.0481$ | $0.77 \pm 0.001$ | $\boxed{0.86 \pm 0.0053}$ |
| | ConvNet | $0.91 \pm 0.0015$ | $0.89 \pm 0.0053$ | $0.95 \pm 0.0064$ | $0.92 \pm 0.0306$ | $0.94 \pm 0.0065$ | $\boxed{0.95 \pm 0.0017}$ | $0.94 \pm 0.0033$ |
| | ResNet | $0.88 \pm 0.015$ | $0.88 \pm 0.0037$ | $0.81 \pm 0.1060$ | $0.93 \pm 0.0110$ | $0.89 \pm 0.0343$ | $0.88 \pm 0.037$ | $\boxed{0.95 \pm 0.0015}$ |
| USPS → MNIST | MLP | $0.51 \pm 0.012$ | $0.42 \pm 0.011$ | $0.46 \pm 0.0031$ | $0.52 \pm 0.0105$ | $0.54 \pm 0.0181$ | $0.57 \pm 0.0072$ | $\boxed{0.62 \pm 0.014}$ |
| | ConvNet | $0.85 \pm 0.0019$ | $0.84 \pm 0.0088$ | $0.90 \pm 0.0720$ | $0.92 \pm 0.0306$ | $0.93 \pm 0.0094$ | $0.91 \pm 0.0052$ | $\boxed{0.95 \pm 0.0064}$ |
| | ResNet | $0.67 \pm 0.11$ | $0.74 \pm 0.031$ | $0.78 \pm 0.0446$ | $0.91 \pm 0.0081$ | $0.89 \pm 0.0125$ | $0.90 \pm 0.0066$ | $\boxed{0.93 \pm 0.007}$ |
| MNIST → MNISTM | ConvNet | $0.53 \pm 0.0064$ | $\boxed{0.68 \pm 0.0081}$ | $0.46 \pm 0.0571$ | $0.65 \pm 0.0149$ | $0.65 \pm 0.0190$ | $0.61 \pm 0.0071$ | $0.49 \pm 0.013$ |
| | ResNet | $0.095 \pm 0.007$ | $0.42 \pm 0.064$ | $0.38 \pm 0.0740$ | $0.58 \pm 0.0249$ | $0.63 \pm 0.0147$ | $0.50 \pm 0.02$ | $\boxed{0.66 \pm 0.01}$ |
| SVHN → MNIST | ResNet | $0.57 \pm 0.11$ | $0.55 \pm 0.12$ | $0.39 \pm 0.1478$ | $0.69 \pm 0.0263$ | $0.70 \pm 0.0220$ | $\boxed{0.78 \pm 0.016}$ | $0.73 \pm 0.0073$ |

# E. Explaining the GS and PM assumptions

In the following, we consider different classifier model classes, different assumptions used in the DA literature, and data settings which relate to the assumptions introduced in our paper. We will need to make two assumptions on the hypothesis class in order to explain the GS and PM assumptions.

**Anti-collapse.** For a given $L_0, \lambda_0 > 0$ pair, we say that a classifier $f$ satisfies *anti-collapse* (AC) for distributions $p$ and $q$, if

$$\Pr_{(x,x') \sim p \otimes q} \left( \ell(f(x), f(x')) \geq L_0 \|x - x'\| \right) > 1 - \lambda_0. \tag{31}$$

Anti-collapse prevents source inputs that are different in some way (e.g., they belong to different classes) from coming too close (depending on $L_0$). One could instantiate anti-collapse to enforce separation of the source domain, which has previously been explored in the literature (Wu et al., 2019).

**Probabilistic Lipschitzness.** Let $f : \mathcal{X} \to \triangle(\mathcal{Y})$ be a classifier and $\ell$ be a metric on $\triangle(\mathcal{Y})$. We say that $f$ is *Probabilistic Lipschitz* (PL) with parameters $(L_1, \lambda_1)$ for distributions $p$ and $q$ if

$$\Pr_{(x,x') \sim p \otimes q} \left( \ell(f(x), f(x')) \leq L_1 \|x - x'\| \right) > 1 - \lambda_1. \tag{32}$$

This definition of PL is a simplification of the Probabilistic Transfer Lipschitzness (PTL) assumption that was defined in (Courty et al., 2017). In particular, we pick the independent coupling between source and target distribution for simplicity. Furthermore, we only look at a single scale $\lambda_1$, rather than requiring a Lipschitz inequality to hold for all Lipschitz constants (with different probabilities). In other words, if PTL holds for the independent coupling (using a specific transfer function $\phi$), then our PL assumption holds for some $\lambda_1$ (our assumption is weaker). It should be noted that (probabilistic) Lipschitzness assumptions are a standard fixture throughout the domain adaptation literature (Ben-David & Urner, 2014; Courty et al., 2017; Wu et al., 2019). Outside of domain adaption, Lipschitzness has been linked to the robustness of models (Bubeck & Sellke, 2021) and several works have explored explicitly constructing Lipschitz neural networks (Wang & Manchester, 2023) or even *bi-Lipschitz* neural networks (Wang et al., 2024). The latter bi-Lipschitz neural network would also enforce the AC assumption we defined above.

We can think of the two assumptions together as probabilistic extensions of *invertibility* (equivalently, bijectivity): injectivity (one-to-oneness) and surjectivity (being onto) are relaxed to hold on the support of the distributions with high probability.

## E.1. Feasibility of Anti-Collapse

In the following section, we discuss why AC is a reasonable assumption to make. To do so, we consider the class of linear models $f(x) = Wx + b$ (in the score / pre-soft-max space) and consider a simplified non-probabilistic version of the anti-collapse assumption. The former simplification to linear models can still be informative: by considering a neural network

with linear score output, the following will apply to the feature space and final layer. The latter simplification is for conciseness of this section. Furthermore, the non-probabilistic version we consider in this section is in-fact a *stronger* assumption than its probabilistic counterpart. Hence, the probabilistic counterpart considered above is actually more realistic than what we will examine here.

Throughout this section, for a linear map $A \colon \mathbb{R}^d \to \mathbb{R}^m$ and a linear subspace $\mathcal{S} \subseteq \mathbb{R}^d$, we write $A|_{\mathcal{S}} \colon \mathcal{S} \to \mathbb{R}^m$ for the restriction of $A$ to $\mathcal{S}$. We define the smallest singular value of $A|_{\mathcal{S}}$ through its variational characterization (i.e., Courant-Fischer Min-Max Theorem (Tao, 2023)):

$$\sigma_{\min}(A|_{\mathcal{S}}) \doteq \inf_{v \in \mathcal{S}, \|v\|_2 = 1} \|Av\|_2 = \inf_{v \in \mathcal{S}, v \neq 0} \frac{\|Av\|_2}{\|v\|_2}.$$

In particular, $\sigma_{\min}(A|_{\mathcal{S}}) > 0$ if and only if $A|_{\mathcal{S}}$ is injective.

**Definition E.1** (Restricted Deterministic Anti-Collapse). Let $\mathcal{V} \subseteq \mathbb{R}^d$ be a set of difference vectors. We say that a representation $f \colon \mathbb{R}^d \to \mathbb{R}^m$ satisfies restricted deterministic anti-collapse on $\mathcal{V}$ with constant $m > 0$ if

$$\|f(x) - f(x')\|_2 \geq m\|x - x'\|_2$$

for every pair $(x, x')$ such that $x - x' \in \mathcal{V}$.

**Proposition E.1.** *Let $f(x) = Ax + b$ for $A \in \mathbb{R}^{m \times d}$ and $b \in \mathbb{R}^m$, and let $\mathcal{V} \subseteq \mathbb{R}^d$ be a set of relevant difference vectors. Define*

$$m_{\mathcal{V}}(A) \doteq \inf_{v \in \mathcal{V}, v \neq 0} \frac{\|Av\|_2}{\|v\|_2}.$$

*If $m_{\mathcal{V}}(A) > 0$, then $f$ satisfies restricted anti-collapse on $\mathcal{V}$ with constant $m_{\mathcal{V}}(A)$.*

*Proof.* For every pair $(x, x')$ such that $x - x' \in \mathcal{V}$, we have

$$\|f(x) - f(x')\|_2 = \|A(x - x')\|_2 \geq \left( \inf_{v \in \mathcal{V}, v \neq 0} \frac{\|Av\|_2}{\|v\|_2} \right) \|x - x'\|_2 = m_{\mathcal{V}}(A)\|x - x'\|_2.$$

As required. $\qquad\square$

**Corollary E.1.** *Let $\mathcal{S} \subseteq \mathbb{R}^d$ be a linear subspace and suppose $\mathcal{V} \subseteq \mathcal{S}$. If the restriction $A|_{\mathcal{S}}$ is injective, then*

$$m_{\mathcal{V}}(A) \geq \sigma_{\min}(A|_{\mathcal{S}}) > 0,$$

*and hence $f$ satisfies restricted anti-collapse on $\mathcal{V}$.*

**Discussion.** There are various reasons why one would expect a "good" linear classifier (i.e., can discriminate between source conditionals) to have weights $W$ which are non-degenerate in the span of these pair sets. First, collapse destroys information: if a linear map sends a nonzero task-relevant direction to zero, then inputs differing in that direction become indistinguishable to any downstream predictor. Second, this viewpoint matches standard principles in dimensionality reduction and statistics, where one seeks to retain informative directions rather than collapse them (e.g., PCA where we want to keep high variance directions). Finally, for overparameterized or highly non-invertible models, global injectivity is neither realistic nor necessary; restricted injectivity on the signal-bearing subspace is the relevant notion, because only those directions need to remain visible for prediction (Bartlett et al., 2020).

### E.2. Relating the assumptions: (GS, PM) to (PL, AC)

We are now in a position to relate the two assumptions GS and PM to the PL and AC assumptions. We start with the PM assumption.

**Probabilistic Margin.** For the probabilistic margin condition, the relevant difference vectors for AC are

$$\mathcal{V}_{\mathrm{pm}} \doteq \left\{ x - x' \mid x \sim p_{\mathrm{x|y}}(\cdot \mid y), x' \sim p_{\mathrm{x|y}}(\cdot \mid y'), y \neq y' \right\}.$$

Thus, it is enough that $W$ be non-degenerate on the span of cross-class source differences. More precisely, if

$$\mathcal{S}_{\text{pm}} = \text{span}(\mathcal{V}_{\text{pm}}) \qquad \text{and} \qquad \sigma_{\min}(W|_{\mathcal{S}_{\text{pm}}}) > 0,$$

then anti-collapse holds on $\mathcal{V}_{\text{pm}}$. Note that the infimum in the definition of our PM assumption has to be relaxed to hold in expectation over the (source, target) samples, unless the classes in the input space are perfectly separable. More generally, one may replace deterministic input-space separation by a probabilistic concentration statement on the product distribution. Namely, if for some $\gamma > 0$ and $\eta \in (0, 1)$ we have

$$\Pr_{\substack{x \sim p_{\text{x}|\text{y}}(\cdot|y) \\ x' \sim p_{\text{x}|\text{y}}(\cdot|y')}} \left( \|x - x'\| \geq \gamma \right) \geq 1 - \eta \tag{33}$$

for every $y \neq y'$, and if anti-collapse holds with parameters $L_0, \lambda_0$, then by an immediate intersection bound,

$$\Pr_{\substack{x \sim p_{\text{x}|\text{y}}(\cdot|y) \\ x' \sim p_{\text{x}|\text{y}}(\cdot|y')}} \left( \ell(f(x), f(x')) \geq L_0\gamma \right) \geq 1 - \eta - \lambda_0.$$

Thus, AC together with probabilistic concentration of cross-class input pairs implies a pairwise PM statement in the output space.

**Gradual Shift.** For the GS assumption, we first consider the decomposition of $q_{\text{x}|\text{y}}$; and then we will consider the condition on the losses afterwards. One can think of each target conditional $q_{\text{x}|\text{y}}$ as being decomposed constructively in the following way:

1. Fix a class label $y \in \mathcal{Y}$ and consider the source conditional $p_{\text{x}|\text{y}}(\cdot \mid y)$ together with the target conditional $q_{\text{x}|\text{y}}(\cdot \mid y)$.

2. Partition the support of $q_{\text{x}|\text{y}}(\cdot \mid y)$ into $s$ sets ordered by increasing distance to the source conditional: $q_{\text{x}|\text{y}}^{(1)}, \ldots, q_{\text{x}|\text{y}}^{(s)}$.

3. Write the target conditional as the mixture $q_{\text{x}|\text{y}}(\cdot \mid y) = \sum_{i=1}^{s} r_i q_{\text{x}|\text{y}}^{(i)}(\cdot \mid y)$, where each $r_i$ is simply the mass of the $i$th partition set under the target conditional. One may then define $a = \min_{i \in [s]} r_i$ and $b = \max_{i \in [s]} r_i$.

4. Let $d_i$ denote an upper bound on the source-to-target component distance for the $i$th piece, meaning that every pair $(x, x') \sim p_{\text{x}|\text{y}}(\cdot \mid y) \otimes q_{\text{x}|\text{y}}^{(i)}(\cdot \mid y)$ satisfies $\|x - x'\| \leq d_i$.

This is constructive: once one pre-specifies $s$ many distances $d_i$ that yield non-empty sets, one obtains a valid decomposition of $q_{\text{x}|\text{y}}$ as required by GS. Having constructed the decomposition, the one-sided GS control follows directly from PL. Indeed, for each component $q_{\text{x}|\text{y}}^{(i)}$, the previous construction guarantees that every pair $(x, x') \sim p_{\text{x}|\text{y}}(\cdot \mid y) \otimes q_{\text{x}|\text{y}}^{(i)}(\cdot \mid y)$ satisfies $\|x - x'\| \leq d_i$. If moreover $f$ satisfies $\text{PL}(L_1, \lambda_1)$ on the same pair distribution, then $\Pr\left( \ell(f(x), f(x')) \leq L_1 d_i \right) \geq 1 - \lambda_1$. Equivalently, denoting $\hat{y} \sim (f_\sharp p_{\text{x}|\text{y}})(\cdot \mid y)$ and $\hat{y}' \sim (f_\sharp q_{\text{x}|\text{y}}^{(i)})(\cdot \mid y)$, we have

$$\Pr\left( \ell(\hat{y}, \hat{y}') \leq L_1 d_i \right) \geq 1 - \lambda_1. \tag{34}$$

Thus, the ordered decomposition in input space induces a *one-sided* probabilistic GS structure in output space, whose indices $i$ can be reordered to yield a *two-sided* GS structure,

$$\Pr\left( (j-1)\varepsilon \leq \ell(\hat{y}, \hat{y}') \leq j\varepsilon \right) \geq 1 - \lambda_1, \ \ a \leq r_j \leq b, \tag{35}$$

with the reordered indices $j = h(i)$ for some mapping $h(\cdot)$. The increment $\varepsilon$ quantifies how gradually the target conditionals move away from the source conditional in output space, as before.

**Remark.** Although the two-sided GS assumption that we employ in the main text can be recovered from (34) by adapting the indices as above, the resulting GS structure could be arbitrarily weak, i.e., $a$ could be arbitrarily small, breaking the GS structure (and the bounds, since they depend on the GS-densities via the $b/a$ term), or requiring a large $\varepsilon$ to describe it. The previously introduced $\text{AC}(L_0, \lambda_0)$ assumption can help to regulate the GS decomposition in that regard. More specifically, assume that the $i$th component is chosen such that every pair $(x, x') \sim p_{\text{x}|\text{y}}(\cdot \mid y) \otimes q_{\text{x}|\text{y}}^{(i)}(\cdot \mid y)$ satisfies

$$d_{i-1} \leq \|x - x'\| \leq d_i,$$

with the convention that $d_0 = 0$. If, in addition to $\mathrm{PL}(L_1, \lambda_1)$, the classifier satisfies $\mathrm{AC}(L_0, \lambda_0)$ on the same pair distribution, then

$$\Pr\left(L_0 d_{i-1} \leq \ell(\hat{y}, \hat{y}') \leq L_1 d_i\right) \geq 1 - \lambda_0 - \lambda_1. \tag{36}$$

The equation (36), as opposed to (34), can help to regulate the densities $r_i$ that appear in any two-sided GS decomposition.

**Remark.** Note that while we used a stronger version of the GS assumption where the band assumption holds *almost surely* due to simplicity,[11] our bounds in Theorem 3.1 and the Corollary 4.1 support the probabilistic extension of the GS assumption considered above. Such an extension only requires that the constants $\lambda_0, \lambda_1$ also appear *multiplicatively* in our bounds, similar to how $\eta$ of the PM assumption appears in the bounds.

### E.3. Anti-Collapse for Nonlinear Models

In this section we give simple sufficient conditions under which the anti-collapse assumption holds for nonlinear models. The cleanest route uses a strong monotonicity condition, which is stronger than a mere lower bound on the smallest singular value of the Jacobian, but has the advantage of implying a global lower Lipschitz bound directly.

**Definition E.2** (Strong Monotonicity). Let $\Omega \subseteq \mathbb{R}^d$ be convex and let $f \colon \Omega \to \mathbb{R}^d$ be a map. We say that $f$ is strongly monotone on $\Omega$ with constant $m > 0$ if

$$\langle f(x) - f(x'), x - x' \rangle \geq m\|x - x'\|_2^2 \qquad \text{for all } x, x' \in \Omega.$$

**Proposition E.2.** *Let $\Omega \subseteq \mathbb{R}^d$ be convex and let $f \colon \Omega \to \mathbb{R}^d$ be strongly monotone on $\Omega$ with constant $m > 0$. Then*

$$\|f(x) - f(x')\|_2 \geq m\|x - x'\|_2 \qquad \text{for all } x, x' \in \Omega.$$

*In particular, $f$ satisfies deterministic anti-collapse on every pair set contained in $\Omega \times \Omega$ with constant $m$.*

*Proof.* By Cauchy–Schwarz and strong monotonicity,

$$\|f(x) - f(x')\|_2 \|x - x'\|_2 \geq \langle f(x) - f(x'), x - x' \rangle \geq m\|x - x'\|_2^2.$$

If $x = x'$, the claim is trivial. Otherwise divide by $\|x - x'\|_2$ to obtain

$$\|f(x) - f(x')\|_2 \geq m\|x - x'\|_2.$$

$\square$

**Corollary E.2** (Jacobian Criterion). *Let $\Omega \subseteq \mathbb{R}^d$ be convex and let $f \colon \Omega \to \mathbb{R}^d$ be continuously differentiable. Suppose there exists $m > 0$ such that for every $z \in \Omega$,*

$$\frac{J_f(z) + J_f^\top(z)}{2} \succeq m I_d, \tag{37}$$

*where $J_f(z)$ denotes the Jacobian matrix of $f$ at $z$. Then $f$ is strongly monotone on $\Omega$ with constant $m$, and hence satisfies deterministic anti-collapse on $\Omega$ with constant $m$.*

*Proof.* Fix $x, x' \in \Omega$ and define $\gamma(t) = x' + t(x - x')$ for $t \in [0, 1]$. Since $\Omega$ is convex, $\gamma(t) \in \Omega$ for all $t$. By the fundamental theorem of calculus,

$$\langle f(x) - f(x'), x - x' \rangle = \left\langle \int_0^1 J_f(\gamma(t))(x - x') \, dt, \, x - x' \right\rangle = \int_0^1 \langle J_f(\gamma(t))(x - x'), x - x' \rangle \, dt.$$

Since only the symmetric part contributes to the quadratic form, we get

$$\langle J_f(\gamma(t))(x - x'), x - x' \rangle = \left\langle \frac{J_f(\gamma(t)) + J_f^\top(\gamma(t))}{2}(x - x'), x - x' \right\rangle \geq m\|x - x'\|_2^2$$

---

[11] Note that one can always make such a reordering, with the caveat being that the density lower bound $a$ could be unnecessarily weak for some conditional distribution $q_{\mathsf{x}|\mathsf{y}}^{(i)}$.

for all $t \in [0, 1]$. Integrating over $t$ yields

$$\langle f(x) - f(x'), x - x' \rangle \geq m\|x - x'\|_2^2.$$

Thus $f$ is strongly monotone, and the conclusion follows from Proposition E.2. □

**Corollary E.3** (Probabilistic Anti-Collapse from a Good Region). *Let $\Omega \subseteq \mathbb{R}^d$ be convex and let $f \colon \Omega \to \mathbb{R}^d$ be continuously differentiable. Suppose* (37) *holds on $\Omega$ with constant $m > 0$. Let $\pi$ be a pair distribution over $(x, x')$ such that*

$$\Pr_{(x,x') \sim \pi} \left( x \in \Omega, \ x' \in \Omega \right) \geq 1 - \eta.$$

*Then*

$$\Pr_{(x,x') \sim \pi} \left( \|f(x) - f(x')\|_2 \geq m\|x - x'\|_2 \right) \geq 1 - \eta.$$

*Proof.* On the event $\{x \in \Omega, \ x' \in \Omega\}$, the conclusion follows from Corollary E.2. Therefore the anti-collapse inequality fails only when $(x, x')$ leaves the good region, which occurs with probability at most $\eta$. □

**Proposition E.3** (Piecewise-Linear Case). *Let $\Omega \subseteq \mathbb{R}^d$ be a convex set and let $f \colon \Omega \to \mathbb{R}^d$ be piecewise affine. Suppose there exists a finite partition*

$$\Omega = \bigcup_{k=1}^{K} \Omega_k$$

*into convex regions such that on each region $\Omega_k$,*

$$f(x) = A_k x + b_k \qquad \text{for all } x \in \Omega_k.$$

*If*

$$\frac{A_k + A_k^\top}{2} \succeq mI_d \qquad \text{for all } k \in [K],$$

*then for every pair $x, x'$ such that the line segment joining them is contained in a single region $\Omega_k$,*

$$\|f(x) - f(x')\|_2 \geq m\|x - x'\|_2.$$

*Proof.* If the line segment joining $x$ and $x'$ lies in $\Omega_k$, then

$$f(x) - f(x') = A_k(x - x').$$

Hence

$$\langle f(x) - f(x'), x - x' \rangle = \langle A_k(x - x'), x - x' \rangle = \left\langle \frac{A_k + A_k^\top}{2}(x - x'), x - x' \right\rangle \geq m\|x - x'\|_2^2.$$

Applying Proposition E.2 to the affine map on $\Omega_k$ yields the claim. □

**Remark.** The nonlinear sufficient condition above is stronger than the linear restricted singular-value condition considered before, but it is simple and fully rigorous. It shows that anti-collapse for nonlinear models follows from a standard non-degeneracy condition on the symmetric part of the Jacobian, or equivalently from strong monotonicity of the representation map on the region visited by the relevant pairs. Finally, we remark that the proposition above can be extended to *probabilistic* anti-collapse similar to Corollary E.3.

### E.4. Alternative Conditions for AC

**Extension to Immersion Mappings.** Note that the previous section considered classifiers whose outputs have the same dimension as the input space. Typically classifiers with high dimensional input data such as images will output predictions in a much lower dimensional output space. That is, in the more general case where $f : \mathbb{R}^D \to \mathbb{R}^d$ with $d < D$, the relevant extension of the above is to consider *immersions* on the data *submanifold* $\mathcal{M}$, with the property that for some $\alpha > 0$

$$v^{\mathrm{T}} J_f(x)^{\mathrm{T}} J_f(x) v \geq \alpha^2 \|v\|^2, \tag{38}$$

for all vectors $v$ in the tangent space to $\mathcal{M}$: $v \in T_x\mathcal{M}$. Written differently

$$\|J_f(x)(x' - x)\| \geq \alpha \|x' - x\|, \tag{39}$$

for nearby points $x, x' \in \mathcal{M}$. The above holds locally whenever the minimum singular value of $J_f$ restricted to the tangent space $T_x\mathcal{M}$ is lower bounded by $\alpha$, i.e., via Courant-Fischer Min-Max Theorem (Tao, 2023) on (38) we have $\sigma_{\min}(J_f|_{T_x\mathcal{M}}) \geq \alpha$ at a given $x$.

**Probabilistic Lipschitzness.** Although the arguments above were restricted to the analysis of the AC assumption for general nonlinear maps $f$, the PL assumption can be shown to hold under similar assumptions, by adjusting the conditions appropriately. That is, the analogue of the above condition $\sigma_{\min}(J_f|_{T_x\mathcal{M}}) \geq \alpha$ is

$$\sigma_{\max}(J_f|_{T_x\mathcal{M}}) \leq L. \tag{40}$$

## F. Diagnostics on the GS and PM assumptions

In this section we provide further diagnostics (beyond the UDA optimization of target accuracy metrics) and evidence regarding the GS and PM assumptions on the standard datasets and neural network models used in Table 1.

### F.1. Evidence regarding the GS assumption

We use target labels to sort the target conditional outputs w.r.t. their distances to the source conditional means. We can do this procedure either separately from the UDA optimization, or during the domain alignment using separate test mini-batches not used for optimizing the network weights. Plots shown in Figures 5 to 9 were generated using test minibatches of size 512. The different colors correspond to the digits 0-9 (10 classes/conditionals in total).

Notice that most conditionals (but not all) are well connected, in terms of the smoothness of the distances, to the source conditionals. Circles and cross markers indicate correctly and incorrectly labeled points, respectively. These markers indirectly indicate the level of margin for each conditional at that mini-batch.

### F.2. Evidence regarding the PM assumption

To validate the PM assumption, we only use source labels from a test mini-batch with 512 samples during the ERM training from scratch (no pretraining). The plots shown below were generated by calculating the 10th percentile (hence the tolerance parameter $\eta = 0.1$) of the empirical distribution of the cross-class conditional distances (i.e., Euclidean distance is the loss). Plots shown in Figures 10 to 12 suggest that the probabilistic margin parameter $M$ grows during the optimization, indicating that the optimizer is able to find good source classifiers with high margin. One exception is for the SVHN case (Figure 12), where the optimizer struggles to separate the data well even for ResNet18, potentially explaining the failure of the $\mathrm{W}^2\mathrm{R}^2$ optimization for that dataset.

Finally, note that such diagnostics can of course be used to construct reliable estimates of the scale $\lambda$ appearing in our bounds. As remarked in the conclusion of the main text, an important research direction for us is to construct reliable estimates of $\lambda$ *without* using target labels. We only note here that our preliminary attempts to do so, using hierarchical clustering or Gaussian mixtures-based modeling, did not yield any benefit, but we still think that a robust estimator is worth pursuing.

## G. Accuracy curves

The accuracy curves, corresponding to Table 1 are shown in Figures 13 - 15 and Figure 6. The test accuracies are shown over the five epochs after pretraining.

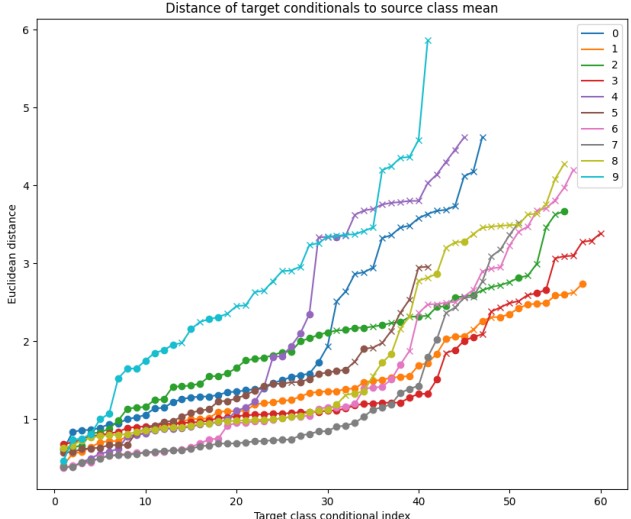

*Figure 5.* Plot indicating gradual shift in the output space of a source-pretrained ResNet18 on the SVHN-MNIST scenario.

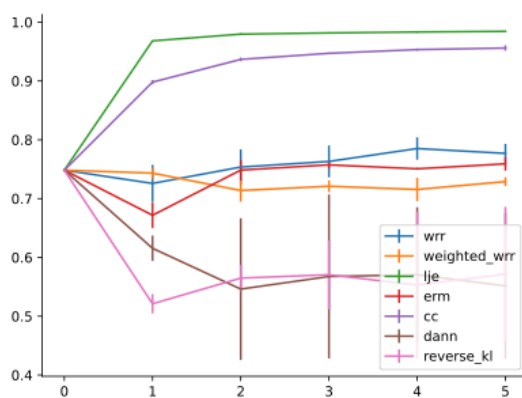

*Figure 6.* Plot indicating test accuracies (over the five epochs after pretraining) of various tested UDA methods using a ResNet18 on the SVHN-MNIST scenario.

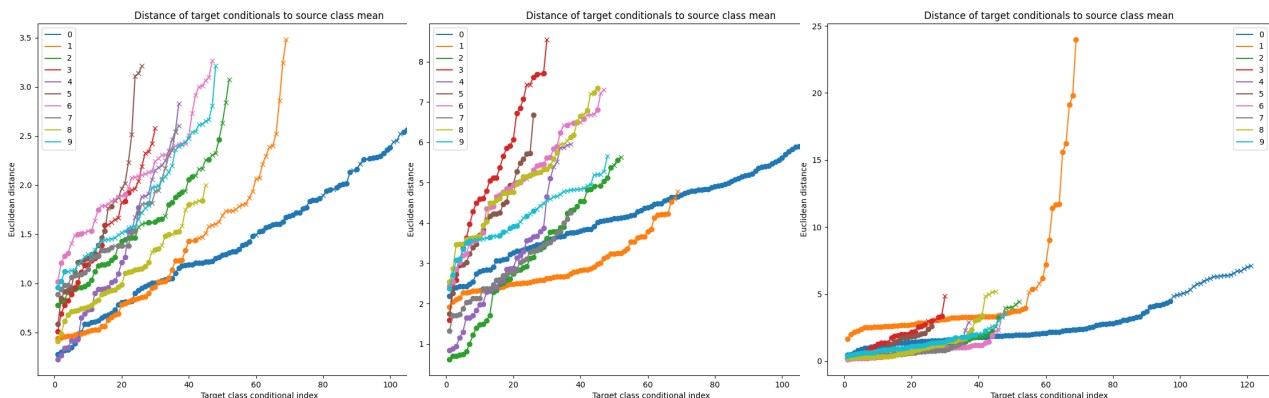

*Figure 7.* From left to right: plots indicating gradual shift in the output space of a source-pretrained MLP, ConvNet and ResNet18 on the MNIST-USPS scenario, respectively.

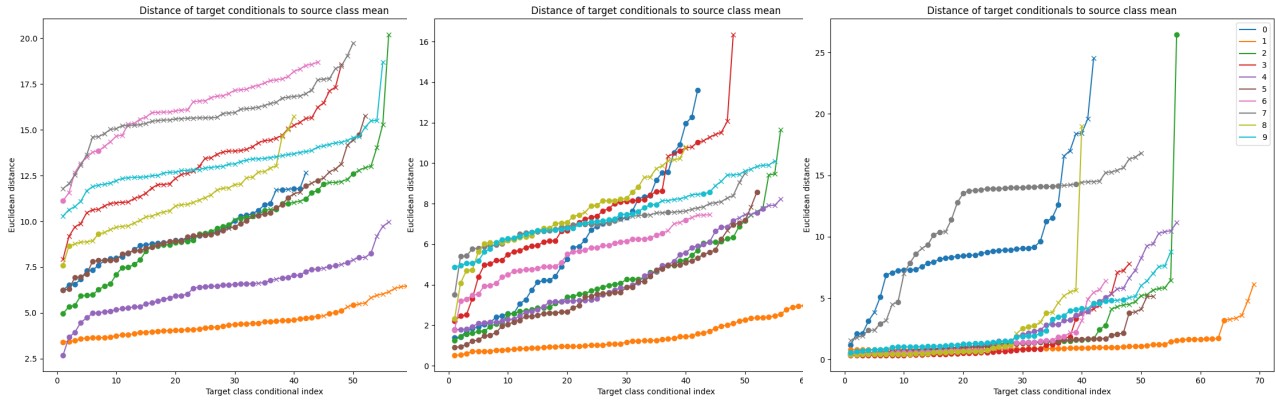

*Figure 8.* From left to right: plots indicating gradual shift in the output space of a source-pretrained MLP, ConvNet and ResNet18 on the USPS-MNIST scenario, respectively.

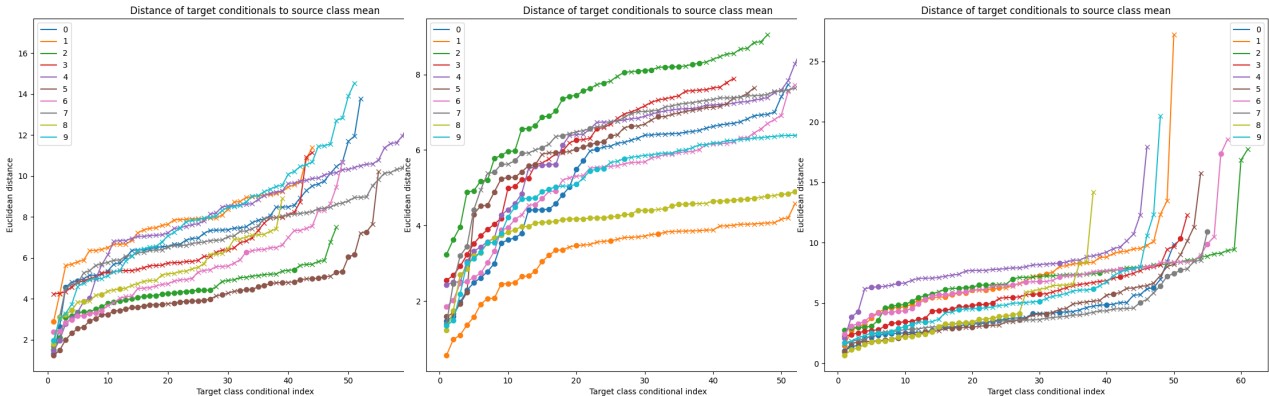

*Figure 9.* From left to right: plots indicating gradual shift in the output space of a source-pretrained MLP, ConvNet and ResNet18 on the MNIST-MNISTM scenario, respectively.

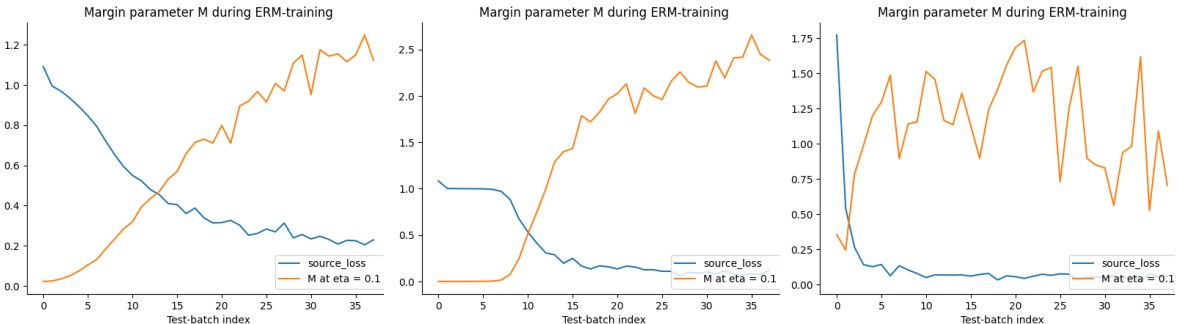

*Figure 10.* From left to right: ERM training on the MNIST dataset of an MLP, ConvNet and ResNet18 model, respectively. Plots show the cross-conditional margin $M$ (with tolerance threshold $\eta$ set to 0.1) increasing as the source loss decreases.

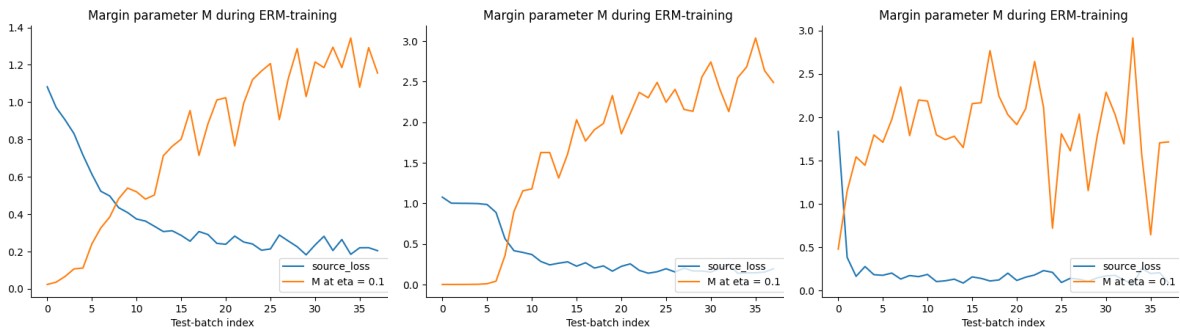

*Figure 11.* From left to right: ERM training on the USPS dataset of an MLP, ConvNet and ResNet18 model, respectively. Plots show the cross-conditional margin $M$ (with tolerance threshold $\eta$ set to 0.1) increasing as the source loss decreases.

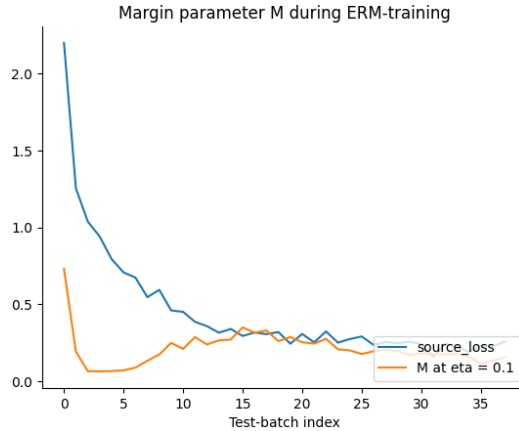

*Figure 12.* ERM training on the SVHN dataset for a ResNet18 model. Plot shows the cross-conditional margin $M$ (with tolerance threshold $\eta$ set to 0.1) as the source loss decreases.

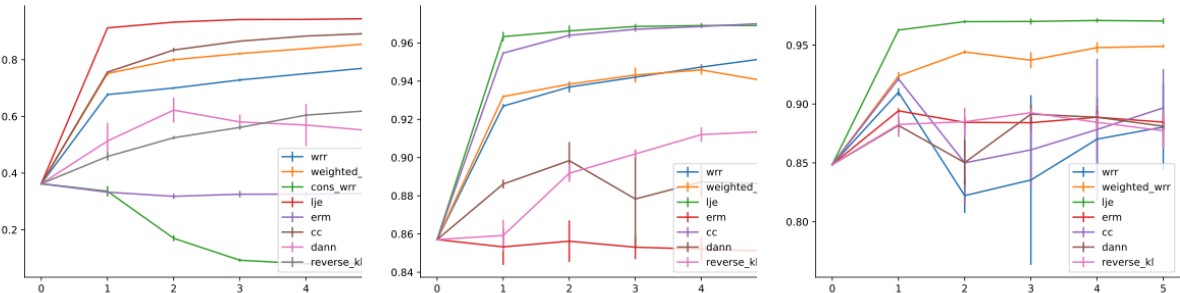

*Figure 13.* From left to right: test accuracy plots (over the five epochs after pretraining) of various tested UDA methods using MLP, ConvNet and ResNet18 on the MNIST-USPS scenario, respectively.

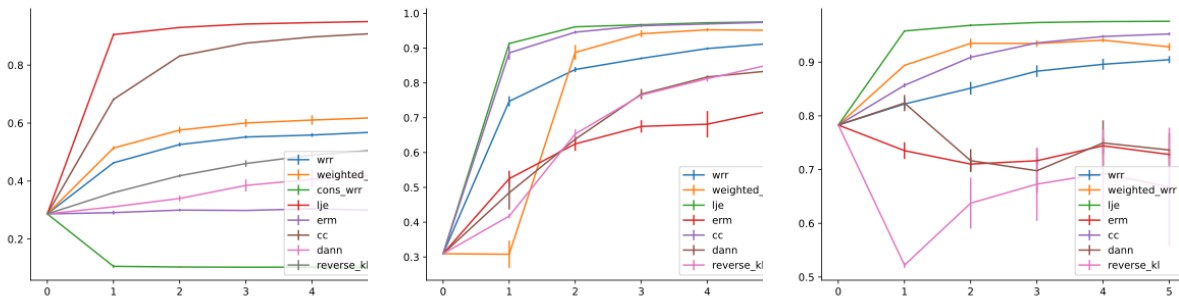

*Figure 14.* From left to right: test accuracy plots (over the five epochs after pretraining) of various tested UDA methods using MLP, ConvNet and ResNet18 on the USPS-MNIST scenario, respectively.

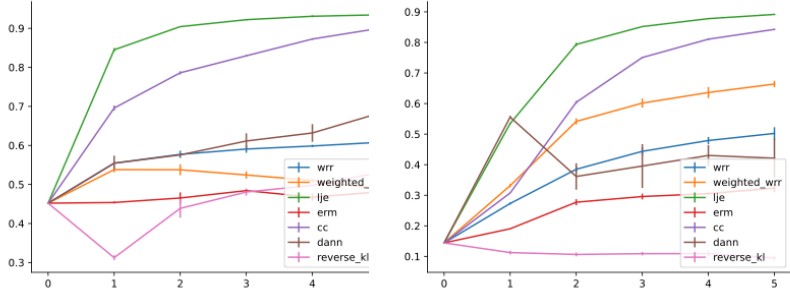

*Figure 15.* From left to right: test accuracy plots (over the five epochs after pretraining) of various tested UDA methods using ConvNet and ResNet18 on the MNIST-MNISTM scenario, respectively.

