# OpenReview forum: "Incorporating Importance Weighting in Optimal Transport Based Domain Alignment"
_ICML.cc/2026/Conference — ICML 2026 regular_

### Official Review · Reviewer_QV5M · 2026-02-21

**Soundness:** 3
**Presentation:** 3
**Significance:** 3
**Originality:** 3
**Overall Recommendation:** 4
**Confidence:** 4

**Summary:**

This work studies the optimal transport (OT) method for the domain adaptation problem, where OT is used to measure the distribution discrepancy. From a theoretical aspect, this work focuses on the intractable terms in existing error bounds and introduces the gradual shift assumption to model the nature of incomputable terms. Under the probabilistic margin assumption, a new error bound is derived, which implies the generalization error can be dominated by the weighted Wasserstein regularization risk. Following the theoretical results, the numerical algorithm of the weighting variant is developed, where the key is to show its equivalence to semi-relaxed OT and unbalanced OT (under entropy regularization). Experiment results are significant compared with other risk objectives.

**Compliance With Llm Reviewing Policy:**

Affirmed.

**Final Justification:**

The authors addressed my concerns in the rebuttal period. Thus, I would like to keep my positive rating and lean toward acceptance.

**Key Questions For Authors:**

Q1. Though the terms in the new error bound are computable, are there any formal guarantees for the feasibility of the PM assumption and the GS assumption in practical scenarios? Besides, the GS assumption seems to rely on the existence of gradual shifting distributions $q^{s}$. Further justifications are appreciated.

Q2. The GS assumption induces additional parameters $(a,b,s,\epsilon)$. How to guarantee the existence of these parameters such that the assumption holds.

Q3. In current experiments, only risk-based objectives are compared. It would benefit from comparison with other OT-based methods for domain adaptation.

Q4. In practical computation, is the regularization coefficient $\lambda$ computable? Specifically, how to determine the parameters that are related to $\lambda$?

**Limitations:**

The authors are encouraged to analyze the potential failure cases of the proposed method, especially the weight estimation and potential computation complexity.

**Strengths And Weaknesses:**

**Soundness**

Pros: The theoretical analysis is reasonable and correct, and the implications are sufficiently discussed. Besides, the theory-inspired model is well connected to the numerical OT algorithm, which ensures the practicality for computation.

Cons: The feasibility of the GS assumption and the PM assumption needs further clarification, and the parameters that determine the assumption also seem to be unobservable.


**Presentation**

Cons: The overall quality is good, where the structure is easy to follow, and the math presentation is clear.



**Significance**

Pros: The new technical content, i.e., refined theory and method under further assumption, is significant. Besides, the experiment results compared with other risk counterparts are also convincing.

Cons: The significance could be further improved by considering a general class of domain adaptation methods, i.e., not only the (regularized) risk objective.



**Originality**

Pros: The theory and method, i.e., weighting regularized OT objective for domain adaptation, are generally novel.

---

> ### Author Rebuttal · Authors · 2026-03-31
>
> We thank the reviewer for the time spent reviewing our paper and the encouraging feedback. Below is our response to the key questions raised.
>
> ---
>
> ### Limitations
>
> > The authors are encouraged to analyze the potential failure cases of the proposed method, especially the weight estimation and potential computation complexity.
>
> In terms of limitations, we will be more explicit in stating them in the next revision. For the weight estimation part, we would like to note that importance weighting in our approach can be both a blessing and curse: although it makes the cost function typically much easier to optimize, it also effects the $\lambda$ term; and without safeguards, can easily force it to go unbounded. This can happen if the semi-relaxed OT selects/weights only a few source points when the target conditionals are shifting from the *unselected* source points). For our work, an unbounded $\lambda$ can cause problems as our theory would not hold in this case. As a result, we modify the original objective to Eq. (30) (as found in the appendix). The idea of this modification (adding unweighted source risk and the UOT regularization) is to introduce safeguards that make sure that the gradually shifting conditionals are indeed shifting from the weighted source distribution with adequate margin (and that the chain of GS distances are not broken, or equivalently, that the GS distance increments $\varepsilon$ are kept small.)
>
> In terms of computational complexity, as mentioned in Appendix D, we use a majorization-minimization based UOT solver [Chapel et al., 2021] (mm-UOT), which is implemented in the POT library [Flamary et al., 2021] to compute the alignment coupling. This is an iterative solver that requires a quadratic amount of computation at each iteration, i.e., $O(Tnm)$, where we remind that $m$ and $n$ are the number of source and target samples, respectively. In practice, we cap the number of iterations $T$ to 1000. The runtime performance of mm-UOT is comparable (albeit a few times slower) to the Sinkhorn-iteration based methods in the POT library.
>
> It should also be noted that in the NN limit, one can just do a single scan of the cost matrix lowering the above complexity to $O(mn)$.
>
> ---
>
> ### Other
>
> > Though the terms in the new error bound are computable, are there any formal guarantees for the feasibility of the PM assumption and the GS assumption in practical scenarios?
>
> Please see our replies to reviewers 1gWR (**Evidence of GS and PM assumptions**) and URHK (**Possible theoretical argument for GS and PM**) on this point. Although there are no formal guarantees we can make at the moment for the feasibility of these assumptions, diagnostics that we provide in the [anonymous link](https://my-random-icml-site.neocities.org/) suggest that the assumptions hold relatively strongly (i.e., with parameters that would keep the scale $\lambda$ not too high) for the UDA datasets and the high-source-accuracy classifiers used in the experiments. One exception seems to be the SVHN case, where the source classifier does not have a high margin, which could help explain the relatively poor results our proposed method obtains in the SVHN-MNIST scenario.
>
> > The GS assumption induces additional parameters. How to guarantee the existence of these parameters such that the assumption holds?
>
> Although there are no guarantees in the general case, plots provided in the link suggest that these parameters would be kept moderate (and hence the scale $\lambda$ would not be too high). As discussed with the other reviewers also, it is possible to rely on other assumptions such as the Probabilistic Lipschitz assumption [Ben-David & Urner, 2014] to provide more insight on this assumption in the practical case. However, fundamentally it remains an assumption: depending on the UDA scenario and the model class, it may be violated (i.e., the gradual shift may be broken).
>
> > In practical computation, is the regularization coefficient computable? Specifically, how to determine the parameters that are related to $\lambda$?
>
> Unfortunately there is in general no way to compute this coefficient without using target labels. Our motivation through the text was to assume that this term would be uniformly bounded. Based on such a bound, minimizing both the source risk and the Wasserstein distance - or its weighted version, which is easier to minimize - would lead to low target risk for moderate values of $\lambda$. We will make sure this point is clear in the main text.
>
> > In current experiments, only risk-based objectives are compared. It would benefit from comparison with other OT-based methods for domain adaptation.
>
> We would like to add more accuracy results in the Appendix, especially the methods that we already discuss there: JDOT, MMD and f-DAL. As for the non-risk-based objectives, do you have any specific methods in mind?

---

> > ### Author Rebuttal · Reviewer_QV5M · 2026-04-01
> >
> > I would like to thank the author for addressing my concerns. Thus, I keep my positive score.

---

### Official Review · Reviewer_raSA · 2026-03-12

**Soundness:** 3
**Presentation:** 3
**Significance:** 2
**Originality:** 3
**Overall Recommendation:** 4
**Confidence:** 3

**Summary:**

The paper studies when minimizing the computable Wasserstein-Regularized Risk (WRR) can actually lead to low target risk in unsupervised domain adaptation. To address the entanglement issue in existing OT-based bounds, the authors introduce the Gradual Shift (GS) and Probabilistic Margin (PM) assumptions and derive a theoretical guarantee for WRR minimization under these conditions. Motivated by optimization difficulties of WRR in practice, they then propose a weighted variant, W2R2, based on source importance weighting, and connect it to semi-relaxed and unbalanced optimal transport. Empirically, the method is evaluated across several benchmark shifts and model classes.

**Compliance With Llm Reviewing Policy:**

Affirmed.

**Final Justification:**

The authors provide more justification of PM and GS assumptions connecting them to anti-collapse and Lipshitzness asm. I raise my score

**Key Questions For Authors:**

1) I found the Gradual Shift assumption somewhat difficult to interpret and stronger/more abstract than the surrounding discussion suggests. My understanding is that it postulates the existence of a decomposition of a target conditional distribution $q_{x|y}$ into a mixture of intermediate conditionals that can be ordered by their distance from the source conditional $p_{x|y}$ in feature space. It is not quite clear how natural or verifiable this assumption is in practice, especially since it depends on the learned representation $f$. A clearer discussion of the practical meaning and limitations of GS would strengthen the paper. Can the authors provide a nontrivial family of models for which GS/PM can be argued to hold? Otherwise, the practical scope of Theorem 3.1 and its relevance to the experimental setting remain somewhat unclear.

2) From my point of view, Figure 2 is not very informative in its current form. It seems to suggest a connected/sequential deformation of the target conditionals, whereas the actual GS assumption only requires the intermediate conditionals to be orderable by their distances from the source conditional in feature space. It would help to replace it with a schematic that explicitly shows (a) the mixture decomposition (b) the role of the distance increments $\varepsilon$, and ideally (c) how GS interacts with PM to control the entanglement term.

3) I found the role of the metric $\ell(\hat{y}, \hat{y}')$ somewhat unclear conceptually. Mathematically, it is a metric on the output space of $f$, and the OT terms / GS / PM assumptions are all formulated in that pushforward space. However, the paper does not always make it explicit whether $f$ should be thought of as a classifier or as a learned representation map, and the practical meaning of distances between $\hat{y}=f(x)$ and $\hat{y}'=f(x')$ therefore remains somewhat under-motivated.

**Limitations:**

yes

**Strengths And Weaknesses:**

### Strengths

1) The paper tackles an important problem in unsupervised domain adaptation by asking when minimizing the computable Wasserstein-Regularized Risk can actually lead to low target risk, which is a meaningful theoretical question for OT-based DA.

2) The weighted WRR / W2R2 idea is interesting; introducing source importance weighting to relax the alignment objective and connecting the resulting formulation to semi-relaxed and unbalanced OT is conceptually appealing.

3) Connecting theory and method: rather than proposing a purely heuristic variant, the authors first motivate WRR through a theoretical bound and then extend the discussion to the weighted case.

### Weaknesses

1) The paper's practical significance hinges to a large extent on the plausibility and practical interpretability of the GS/PM assumptions. I found the idea in Section 4 quite interesting, especially the connection between importance weighting and semi-relaxed / unbalanced OT. However, this part of the paper builds heavily on the theoretical setup of Section 3, and I still have a number of questions about the meaning and practical plausibility of the GS/PM assumptions (see below).

2) The presentation could also be improved. The paper is at times difficult to follow because it requires frequent back-and-forth with the notation, and defining key objects closer to their first appearance would make the exposition clearer.

---

> ### Author Rebuttal · Authors · 2026-03-31
>
> We thank the reviewer for the time invested in reviewing our paper and we look forward to a fruitful discussion.
>
> > The paper is at times difficult to follow because it requires frequent back-and-forth with the notation, and defining key objects closer to their first appearance would make the exposition clearer.
>
> Due to the page limit, we were forced to concentrate the technical prerequisites in the Background and Related Work section. But we agree that defining the key objects closer to their first appearance would make the paper more readable. Do you have any specific examples in mind? We think for instance that the UOT definition could be moved to Section 4 where it's first needed.
>
> > I found the Gradual Shift assumption somewhat difficult to interpret and stronger/more abstract than the surrounding discussion suggests .... A clearer discussion of the practical meaning and limitations of GS would strengthen the paper. Can the authors provide a nontrivial family of models for which GS/PM can be argued to hold? Otherwise, the practical scope of Theorem 3.1 and its relevance to the experimental setting remain somewhat unclear.
>
> Please find our discussion of this topic in the response to reviewer 1gWR, under the title **Evidence of GS and PM Assumptions**. To help elucidate the practical meaning of our assumptions, we have added a number of diagnostic empirical plots in the following [anonymous link](https://my-random-icml-site.neocities.org/). The plots suggest that target conditionals shift gradually (in the output space and based on the scalar distance defined by the metric loss) for various high-source accuracy (ERM-trained) neural networks and UDA datasets. As for the PM assumption, the margin parameter $M$ consistently increases for most datasets as the source risk decreases. Interestingly, the margin is relatively low for the SVHN classifier, which could help to explain the relatively poor results on that dataset, as our proposed method has increased sensitivity to the margin (as opposed to the standard WRR).
>
> > From my point of view, Figure 2 is not very informative in its current form .... It would help to replace it with a schematic that explicitly shows (a) the mixture decomposition (b) the role of the distance increments, and ideally (c) how GS interacts with PM to control the entanglement term.
>
> We agree with the reviewer that the figure could be improved. As remarked in the main text: *Note that unlike what the illustration suggests, the shifting distributions do not have to be connected ...* but will make additional modifications to address your concerns. As for (c), we think that Figure 4 plays that role better, but we couldn't include it in the main text due to space constraints. We included a modified version in the [anonymous link](https://my-random-icml-site.neocities.org/) to make the visualization more informative and we will see if we have space to include it in the main text.
>
> > The paper does not always make it explicit whether $f$ should be thought of as a classifier or as a learned representation map
>
> In the experiments we align the source and target distributions in the *output* space, and the notation was meant to be confined to that space. We will make sure this is clear in the main text.

---

> > ### Author Rebuttal · Reviewer_raSA · 2026-04-04
> >
> > Dear authors, thank you for responses.
> > My main concern is still unresolved: I believe, that plots are not enough to support the claim of the main theorem. Therefore, I would ask for providing at least 2-3 nontrivial families of models for which GS/PM holds together with rigorous proof and experimental validation. Otherwise, in my view, the value of the theoretical result remains limited.

---

> > > ### Author Response · Authors · 2026-04-08
> > >
> > > Dear reviewer, thank you for your continued interest in our paper and we are sorry that it took us so long to reply. We took some time to construct a proper reply that can hopefully answer some of your concerns. We have added a detailed formal argument on how our assumptions (GS and PM) connect to various other assumptions commonly utilized in the literature, e.g., different variants of Lipschitzness and another assumption that we called anti-collapse in the document.
> > > These details can be found in the left sidebar of the previous link used (also relinked here: [anonymous link](https://my-random-icml-site.neocities.org/anticollapse)).
> > >
> > > As the reviewer suggested, ideally one would like to prove that certain model families ensure that our assumptions hold. Unfortunately, the story is more complicated as our GS and PM assumptions are data dependent, but we hope that the included formal arguments elucidate how our assumptions connect to previously explored assumptions and properties that ``good'' models are assumed to hold. Note that the domain adaptation literature similarly employs such data dependent assumptions. For instance, please see Wu et al., 2019's combination of regular Lipschitzness and their data dependent alignment and separation assumptions; or consider Courty et al., 2017's usage of probabilistic transfer Lipschitzness, which involves couplings over the source and target distributions as well as the model.

---

### Official Review · Reviewer_URHK · 2026-03-12

**Soundness:** 3
**Presentation:** 2
**Significance:** 3
**Originality:** 3
**Overall Recommendation:** 5
**Confidence:** 3

**Summary:**

The paper presents a twofold contribution to the domain adaptation literature.

First, the authors show that under suitable assumptions (called Gradual Shift and Probabilistic Margin) on the source and domain distribution of a domain adaptation task, the optimal transport (OT) objective known as Wasserstein Regularized Risk (WRR) effectively provides an upper bound to the empirical risk associated to the target distribution (when, in general, the WRR only represents the computable part of an upper bound on the empirical risk).

Second, the authors propose to incorporate an importance weighting strategy into WRR, yielding the Weighted Wasserstein Regularized Risk (W²RR) objective, and they show that the minimization of W²RR corresponds to (depending on the regularization) a semi-relaxed or an unbalanced optimal transport problem. They then show through numerical experiments on unsupervised domain adaptation tasks that W²RR outperforms various alternative approaches.

**Compliance With Llm Reviewing Policy:**

Affirmed.

**Final Justification:**

The main strengths of the paper are that it presents an original theoretical argument explaining why the WRR objective works in practice in unsupervised domain adaptation settings, and proposes an enhancement of the WRR using an importance weighting strategy. Broadly speaking, my main concerns were of two types: (1) presentation issues about some results and (2) the interpretation and plausibility of the proposed GS and PM assumptions in concrete settings. However, the authors' rebuttal addressed them satisfactorily.

**Key Questions For Authors:**

1. You state at the end of Section 3 that the GS and PM parameters need to hold uniformly for the class of classifier functions that is being explored during optimization. Could you clarify what you meant in the following passage, which I found a bit opaque: "the sequence of models $f_k$ need to have enough regularity such that they do not hide the shift at some intermediate $k$ [...] this can happen if, e.g., the optimized network resorts to using spurious features exclusively" ?

2. Are there any theoretical insights on when we can expect a pair of distributions $(p,q)$ and a set of classifiers $\mathcal{F}$ to satisfy a GS and PM assumption with uniform parameters ?

3. In practice, how can one estimate the GS and PM parameters corresponding to a given source and target distributions and a class of classifiers, and determine the value to give to the parameter $\lambda$ for the bound in Theorem 3.1 to hold ?

4. Could the authors clarify which of their results require the loss $\ell$ to be a metric ? For instance, does Theorem 3.1 require it ? The assumption is not present in its statement but its proof uses Lemma A.3 which itself uses the metricity of $\ell$. If Theorem 3.1 requires $\ell$ to be a metric, can it be relaxed to the case where $\ell$ satisfies only a relaxed triangle inequality (as in Koç et al 2025, Appendix D), as the first paragraphs of Appendix B seem to suggest ? Still on the assumptions of Theorem 3.1, its restatement in Appendix A additionally requires $\ell$ to be bounded from above and below, but the proof does not seem to use this. Could you address this inconsistency ?

**Limitations:**

Yes.

**Strengths And Weaknesses:**

To the best of my knowledge, the paper's contributions are novel and significant, and the proofs and reasonings are sound (I have checked the proofs in Appendix A but I have not checked in too much detail the developments in Appendix B).

On the other hand, there are some issues in the paper's overall presentation. The content of the assumptions as well as the assumptions of the theorems are sometimes not completely clear ; in particular, the statements of the GS and PM assumptions should clearly state that they hold for a fixed classifier $f$ (see also the key questions below). The notations used to denote the various conditionals can at times be confusing and are not always consistent (for instance the pushforward of a joint distribution $p$ by $(f,id)$ is denoted $f$#$p$ in the Notations section, but it is often denoted $f$#$p_{X|Y}$ in the rest of the paper).

Some minor comments/remarks:
- In the Notations section (line 107), the $\mathbb{Y}$ should be replaced by $\mathcal{Y}$
- It should be mentioned somewhere that $\mathcal{H}$ denotes the space of classifiers
- In Appendix A, line 658: $\mathrm{supp}(p_Y | y')$ should be $\mathrm{supp}(p_Y)$
- In lines 662-667: should the $\min$ not actually be an $\inf$ ?
- Line 675: $p_{X|Y}^{(i)}$ should be $p_{X|Y}$ (under the first "Pr")
- Lines 763-764: $\hat{y}'$ should be quantified over $\mathrm{supp}(fp_X)$. The line "Taking $\hat{y}'' \in argmin_{\hat{y}} \ell(\hat{y},\hat{y}')$ should be replaced by "Taking $\hat{y}' \in argmin_{\hat{y}'} \ell(\hat{y}'',\hat{y}')$
- In Corollary A.2 and its proof, shouldn't the $1-\eta$ factors be replaced by $1-\eta|\mathcal{Y}|$ ?
- At lines 992-996 there is missing a $1/m$ factor in the $X_p^TX_p$ term in the definition of $H$, and in the right hand side of the equation defining $\beta^\star$. Also $\beta$ should be replaced by $\beta^\star$

---

> ### Author Rebuttal · Authors · 2026-03-31
>
> We would like to thank the reviewer for the detailed review and the encouraging feedback. Below is our response to the key questions raised.
>
> ---
>
> ### Possible theoretical arguments for GS and PM
>
> > Are there any theoretical insights on when we can expect a pair of distributions $p, q$ and a set of classifiers $\mathcal{F}$ to satisfy a GS and PM assumption with uniform parameters.
>
> In general, uniform bounds on the parameters (and as a result on $\lambda$) would be hard to derive for highly non-invertible classifiers such as DNNs. However, we mentioned in the experiment section in the Appendix that CNNs are claimed to be *more invertible* than nonconvolutional ones [Gilbert et al., 2017], which appears to help with the UDA optimization. We think that this argument can be made more rigorous using *Probabilistic Lipschitzness* (PL) [Ben-David & Urner, 2014]. Even if the function classes optimized in practice (DNNs) are not uniformly smooth and Lipschitz (with high probability over the data), it would be enough to assume PL over *Rashomon sets* $H_r = \lbrace f | \epsilon_p(f) \leq r \rbrace$ and use localization techniques [Wang & Mao, 2024] to derive uniform bounds. This would make rigorous our argument in the main text where we noted that deep neural networks (DNNs), although notoriously good at picking up spurious correlations in the data [Geirhos et al., 2020], have to resort to some invariant latent features to separate the source class conditionals accurately.
>
> To explain the above more intuitively, *invariant latent features* assumes that a GS assumption occurs in some latent space, which via PL is necessarily *revealed* in the output or representation space of a high-source-accuracy classifier.
> Our diagnostic plots in the [anonymous link](https://my-random-icml-site.neocities.org/) suggest that such a theoretical bound would not be unrealistic.
>
> Finally, if we assume that the source and target distributions are Gaussian, it is possible to refine the arguments sketched out in the Gradual Shift subsection of Appendix B for linear classifiers to derive a uniform bound on $\lambda$.
>
> ---
>
> ### Other
>
> > The notations used to denote the various conditionals can at times be confusing and are not always consistent.
>
> Thank you for the suggested corrections, we will incorporate them. As for the notation used to denote the pushforward of a *joint* distribution $f  {\sharp} p$, it is actually not used in the current manuscript, so we will remove it to avoid any confusion.
>
> > Could you clarify what you meant in the following passage, which I found a bit opaque: "the sequence of models $f_k$ need to have enough regularity such that they do not hide the shift at some intermediate $k$ [...] this can happen if, e.g., the optimized network resorts to using spurious features exclusively"?
>
> We realize that *hiding* the shift is a confusing explanation. We meant to say that the intensity of gradual shift (and hence the actual values of the GS parameters) in the output space $f {\sharp} p$ depends on the model $f$ and that during the optimization of the weights, this *shift intensity* can be weakened (i.e., the conditionals become significantly distant, as will inevitably happen for a high-source-accuracy but low-target-accuracy classifier). This would show itself in the bounds as an inflation of $\lambda$.
>
> > In practice, how can one estimate the GS and PM parameters corresponding to a given source and target distributions and a class of classifiers, and determine the value to give to the parameter $\lambda$ for the bound in Theorem 3.1 to hold ?
>
> As we discussed in the reply to reviewer 1gWR's question (**Evidence of GS and PM assumptions**), in practice estimating such parameters requires access to target labels. It may be possible to come up with a method that upper bounds the parameter $\lambda$, however our preliminary experiments did not suggest an easy way to do so.
>
> > Could the authors clarify which of their results require the loss to be a metric ? ... If Theorem 3.1 requires $\ell$ to be a metric, can it be relaxed to the case where $\ell$ satisfies only a relaxed triangle inequality (as in Koç et al 2025, Appendix D), as the first paragraphs of Appendix B seem to suggest ? Still on the assumptions of Theorem 3.1, its restatement in Appendix A additionally requires to be bounded from above and below, but the proof does not seem to use this. Could you address this inconsistency ?
>
> We assume that the loss function is a metric throughout the main text. We realize that we omitted to mention this in Theorem 3.1, we now added it. The assumption can indeed be relaxed the same way as in Koç et al 2025 and we used such a relaxation in Appendix B. We will note this in the main text and connect better to the Appendix.
>
> Finally, the restatement in Appendix A was inconsistent with the original statement, we corrected it. Thank you for bringing these points to our attention!

---

> > ### Author Rebuttal · Reviewer_URHK · 2026-04-03
> >
> > I thank the authors for their detailed reply. Since I believe that my questions have been adequately answered to, I will raise my score from "weak accept" to "accept".

---

### Official Review · Reviewer_1gWR · 2026-03-13

**Soundness:** 3
**Presentation:** 3
**Significance:** 3
**Originality:** 3
**Overall Recommendation:** 4
**Confidence:** 4

**Summary:**

This paper studies unsupervised domain adaptation from an optimal transport perspective. The paper starts from Wasserstein-regularized risk and argues that, under two assumptions called gradual shift and probabilistic margin, better optimization of the computable part of the bound can translate into lower target risk. Motivated by the observation that the standard Wasserstein-regularized objective can be difficult to optimize, the paper introduces a weighted variant, W2R2, which applies importance weighting to the source distribution. The paper further shows that this weighted formulation is connected to semi-relaxed optimal transport and, with additional regularization, to unbalanced optimal transport. Empirically, the method is evaluated on several standard digit-domain adaptation benchmarks across multiple model classes, where it often improves over prior OT-based or adversarial baselines and is claimed to be relatively robust across shifts without retuning.

**Compliance With Llm Reviewing Policy:**

Affirmed.

**Final Justification:**

The rebuttal and follow-up discussions significantly clarified the GS/PM assumptions and the novelty of the Wasserstein-based risk bounds. While some gaps between the theoretical framework and practical implementation persist, the work offers a solid and valuable foundation. Given the overall positive consensus and the authors' professional engagement, I am happy to support a Weak Accept. I encourage the authors to further bridge the theory-practice gap in future iterations.

**Key Questions For Authors:**

1. Can the authors provide results on at least one broader non-digit UDA benchmark under the same no-retuning-across-scenarios protocol? If the gains persist there, I would view the significance of the paper more favorably.

2. The theorem suggests a regularization strength that depends on quantities tied to the GS and PM assumptions, but the experiments use a fixed lambda and then optimize a modified practical objective. Can the authors clarify this theory-to-practice gap and provide ablations isolating the effect of each modification? A convincing answer would improve my view of soundness.

3. The paper argues that the empirical results support the GS and PM assumptions. Can the authors provide direct empirical proxies or diagnostics for these assumptions, beyond final target accuracy? That would make the theoretical story much more convincing.

4. Can the authors explain the failure cases in Table 1, especially MNIST to MNISTM with ConvNet and SVHN to MNIST with ResNet, and characterize when the weighting hurts rather than helps? A clear answer would improve my confidence in the robustness claim.

5. Can the authors include the full comparison against the additional baselines discussed in the appendix under the same fixed-hyperparameter setting?

**Limitations:**

The paper does not adequately discuss limitations or potential negative impact. The impact statement is essentially empty. At minimum, the authors should discuss: (i) when the GS and PM assumptions are likely to fail, (ii) the dependence on pretraining and solver choices, and (iii) the risk that domain adaptation under poorly characterized shift can create false confidence or uneven performance across target subpopulations.

**Strengths And Weaknesses:**

This paper studies optimal-transport-based unsupervised domain adaptation from both a theoretical and an algorithmic angle. The most interesting part for me is the move from standard Wasserstein-regularized risk to a weighted version, W2R2, and the observation that this induces a semi-relaxed OT / UOT view rather than exact matching. That is a useful perspective: when source and target already overlap, exact alignment can be unnecessarily restrictive, and the paper gives a clean intuition for why a weighted relaxation may help. I also found the appendix helpful in motivating the method, since it shows that plain WRR can already be non-convex and can have poor local optima even in simple linear settings.

On soundness, the theory looks mostly careful, and the appendix contains substantial proof detail. The paper explicitly states the GS and PM assumptions and derives a bound under which minimizing the computable part of the objective can control target risk. Still, these assumptions are fairly strong, and the practical method does not estimate or verify the quantities that determine the recommended regularization strength in the theorem. In experiments, the method uses a fixed lambda and then optimizes a modified UOT-style objective described more fully in the appendix. Because of that, I see the theory more as a plausible explanation for why this family of methods may work, rather than a tight justification of the exact training recipe used in practice.

The empirical results are promising, but the support is limited. Table 1 shows several strong wins, for example on MNIST to USPS with MLP and on USPS to MNIST with ConvNet. At the same time, the method is not uniformly best: on MNIST to MNISTM with ConvNet, W2R2 is clearly worse than both DANN and WRR, and on SVHN to MNIST with ResNet it is worse than WRR. More broadly, the evaluation is still confined to digit adaptation benchmarks across four transfers and three model classes, with ten training epochs and five runs. That makes the “state-of-the-art” and “robust without hyperparameter tuning” claims feel stronger than the evidence currently supports.

The appendix also weakens the robustness story a bit. Some important practical details are deferred there, including the exact UOT variant and optimization choices. More generally, the method seems to depend on specific regularization and pretraining choices to behave well, which suggests that the practical stability story is somewhat more qualified than the main text suggests.

Presentation is decent overall. The paper is readable, the figures are helpful, and the related-work discussion is fairly honest about how this paper extends earlier OT, UOT, and weighting-based alignment ideas. My main presentation issue is that several key practical details are deferred to the appendix, including the exact objective used in experiments and several important training details. The main paper would benefit from a short summary of what is actually optimized in practice, and from a clearer separation between the theorem-motivated objective and the final implemented one.

In terms of significance and originality, I do think there is real value here. The weighted-WRR to semi-relaxed OT/UOT connection is interesting, and the paper gives a useful way to think about relaxed matching when the two domains share support. At the same time, the paper itself positions the contribution as an extension of recent OT-based DA work, and the empirical scope is too narrow for me to view the broader impact as established yet.

---

> ### Author Rebuttal · Authors · 2026-03-31
>
> We would like to thank the reviewer for the careful review and the encouraging feedback. Below is our response to the key questions raised.
>
> ---
>
> ### Evidence of GS and PM assumptions
>
> > The paper argues that the empirical results support the GS and PM assumptions. Can the authors provide direct empirical proxies or diagnostics for these assumptions, beyond final target accuracy?
>
> We agree that GS and PM assumptions seem strong. We are happy to provide more detail on their feasibility, hopefully the additional empirical evidence will help interpret our positive results in the Experiments section. Please find the additional figures in the [anonymous link](https://my-random-icml-site.neocities.org/), where the plots show that these assumptions do hold reasonably well (meaning that $\lambda$ *can* be kept moderately small). In the link we included plots for various datasets and source-trained models. In the first set of plots, we show the inter-class conditional distances (GS) and for the second set of plots, we show the 10% percentile cross-class distance lower bounds (PM). We will update the paper to include a subsection on these diagnostics, thank you.
>
> It is also true that *the practical method does not estimate or verify the quantities that determine the recommended regularization strength in the theorem.*
> It is not critical to know the precise value of the $\lambda$ coefficient appearing in the bound, but estimating an upper bound could make our proposed algorithm more cautious, and help avoid the failure cases. Unfortunately, figuring out a good scale requires access to the target labels or to the data generative mechanism. We tried various heuristics based on pseudolabeling (e.g., hierarchical clustering, gaussian-mixtures based modeling) to estimate the GS intensity but the methods we tried failed to provide any benefit. As for PM, estimating its parameters only requires access to source labels and hence is much more feasible.
>
> However, we still think that it may be possible to estimate the GS intensity (and hence its parameters) in a robust way in certain cases, without access to target labels. Together with the PM parameter estimates, one could then construct an upper bound on $\lambda$, which could moreover - due to its dependence on the model $f$ - be auto-differentiated. This would be out of scope for the current paper, but we will mention these extensions as an important future research direction.
>
> ---
>
> ### Other
>
> > The theorem suggests a regularization strength that depends on quantities tied to the GS and PM assumptions, but the experiments use a fixed lambda and then optimize a modified practical objective. Can the authors clarify this theory-to-practice gap and provide ablations isolating the effect of each modification?
>
> Our theory suggests that the success of the experiments hinges strongly on whether the $\lambda$ term can be kept bounded: as long as this term is bounded (with a moderate value), then a successful WRR minimization (with $\lambda = 1$) will force target risk to be small, even though we use a different scale. However, as discussed in the manuscript, WRR is often difficult to optimize effectively, and hence we introduced importance weighting to make it easier to optimize.
>
> The reviewer may also want to look at **Limitations** in reviewer QV5M's reply section, where we clarify some other practical changes made.
>
> > Can the authors explain the failure cases in Table 1, especially MNIST to MNISTM with ConvNet and SVHN to MNIST with ResNet, and characterize when the weighting hurts rather than helps?
>
> This question connects well with the points above. The failure cases stem from the eventual breaking of the GS and PM assumptions. Our added plots hopefully illustrate this clearly. In particular, the margin SVHN plot indicates failure of the PM-assumption (for the specific percentile).
>
> > Can the authors include the full comparison against the additional baselines discussed in the appendix under the same fixed-hyperparameter setting?
>
> We realize that these results were only discussed qualitatively (e.g., "does not work better than", or "is not stable" etc.) in the Appendix and that in general the additional discussion in the Appendix was not integrated well with the main text. We will extend the results given in Table 1 to include the other algorithms. We will also extend the discussion in the main text to include the exact objective used in the experiments and the various training details.
>
> > The paper does not adequately discuss limitations or potential negative impact.
>
> To the best of our understanding, the impact statement's intent is to focus on the more general ethical and societal consequences of the work. Nevertheless, we agree that we could have been clearer on stating the limitations of our work and will add a dedicated paragraph near the conclusion to include your stated points.
>
> We will also add other limitations discussed in reviewer QV5M's **Limitation** section.

---

> > ### Author Rebuttal · Reviewer_1gWR · 2026-04-04
> >
> > Thank you for the rebuttal. The response helps clarify several points, especially the intended role of the GS/PM assumptions, the explanation of some failure cases, and the authors’ plan to add more diagnostics and fuller baseline comparisons. That said, my main concerns are not fully resolved at this stage. In particular, the gap between the theory and the practical training objective remains only partially clarified, and the broader empirical support is still limited in the current submission. Several points are addressed mainly by planned revisions or additional discussion, rather than by substantial new evidence in the paper itself. I therefore keep my original score unchanged.

---

> > > ### Author Response · Authors · 2026-04-08
> > >
> > > Dear reviewer, thank you for your response to our paper. We agree with your raised points and due to time constraints we concentrated on a detailed response regarding the practicality of our assumptions: in particular, how our assumptions connect to previously explored assumptions in the literature. Please find our response located in an additional left side bar of the previous [anonymous link](https://my-random-icml-site.neocities.org/anticollapse). There are also some additional discussion points highlighted in the reply thread to reviewer raSA.
> > >
> > > On the specific angle of practicality, we show in the latest reponse why certain assumptions that imply versions of our own assumptions (GS and PM) would be expected in good performing models. We hope that these additional theoretical connections complement our previous response / empirical arguments. Although our assumptions have failure modes for certain model + dataset combinations, which we tried to mitigate through various algorithmic choices (i.e., those that you highlighted in the Appendix), we believe that our result showing that
> > >
> > > (i) the target risk can be directly bounded by a scalar multiple of the Wasserstein regularized source risk without any additional terms (as far as we know, such a bound does not exist in the literature) and that
> > >
> > > (ii) such a bound can be tightened by importance weighting (although with implications for the scale of the bound as highlighted in Corollary 4.1 and in the experiments)
> > >
> > > is a useful starting point from which the specific assumptions used as well as the algorithm can be refined in future work.

---

### Decision · Program_Chairs · 2026-04-30

**Decision:**

Accept (regular)

**Comment:**

There is a consensus among the reviewers that the contribution is original and significant. The authors' rebuttal properly addressed initial concerns about the validity of the assumptions.

It is important that the authors include the clarifications and new results from their rebuttal in the final manuscript.